# Functional analysis within latent states: A novel framework for analysing functional time series data

**Owen Forbes**[ID]*, **Edgar Santos-Fernandez, Paul Pao-Yen Wu**[ID]**, Kerrie Mengersen**

QUT Centre for Data Science, School of Mathematical Sciences, Queensland University of Technology, Brisbane, QLD, Australia

* owen.forbes@hdr.qut.edu.au

## Abstract

Functional data analysis (FDA) enables modelling and interpretation of data represented as functions over a continuum like time, space, or frequency. This paper introduces the *flawless* analysis framework (FunctionaL Analysis Within LatEnt StateS), a nested FDA framework for analysing functional time series data. It provides comprehensive insights into the interplay between latent state characteristics, state occupancy dynamics, and functional attributes within states, while maintaining interpretability at each level. Applying *flawless* to functional time series of power spectral densities from electroencephalography (EEG) data from the Healthy Brain Network, we explore functional characteristics of resting state brain activity in $n = 503$ early adolescents aged 9 - 15 ($\bar{X} = 11.5$, $SD = 1.7$). We identify four functional latent states associated with variations in psychopathology and cognitive function. Bayesian regression models reveal important associations between the dynamics of latent state occupancy, functional traits within states, and relevant health measures. The integration of multiple FDA tools offers rich insights into functional and time-frequency characteristics of longitudinal data. For neuroscientific data this requires fewer assumptions about oscillatory peak frequencies, and captures more detailed frequency domain characteristics. *flawless* offers utility for novel and sophisticated insights into functional time series data across a range of areas for research and practice.

## Introduction

Functional data analysis (FDA) is a growing field of statistical research regarding analysis of data that can be represented as functions, including curves or shapes which vary over a continuum such as time or frequency [1–3]. It is a rapidly expanding area of research and application within medical statistics, and has broad utility across applications from genetics to neuroscience and prediction of clinical outcome time series for health services [4–6]. FDA involves analysing data that are intrinsically infinite dimensional, which poses challenges for both theoretical development and computational methods. However, this inherent high dimensionality also provides a rich source of information and many opportunities for research and data analysis. Sequences of functional data observed over time are an increasingly common subject of applied statistical analysis across a variety of fields. This flavour of functional data occurs

data used for this work are available through the Child Mind Institute's Healthy Brain Network Scientific Data Portal, accessible at https://fcon_1000.projects.nitrc.org/indi/cmi_healthy_brain_network/. Data used for this study were contained in Release Numbers 1.1 - 10 of HBN, with Release Dates between January 31, 2018 and April 13, 2022.

**Funding:** Full list of funders: Australian Research Council Centre of Excellence for Mathematical and Statistical Frontiers, CE140100049, KM Statistical Society of Australia, PhD Scholarship, OF Queensland University of Technology, PhD Scholarship, OF International Biometrics Society, PhD Scholarship, OF.

**Competing interests:** The authors have declared that no competing interests exist.

commonly across domains such as ecology, climatology and biology [7]. Examples of functional data observed over time in these domains include animal movement data on orca dives [8], biomechanical data from athletes [9,10], and power spectral density curves measuring frequency content of brain activity over time [11].

In neuroscience, functional data are common objects of investigation, with several studies demonstrating FDA's utility for analysing brain activity data varying over the functional domains of time, space and frequency [11–13]. For example, Hasenstab et al. developed a multi-dimensional FPCA approach for EEG analysis, while Scheffler et al. extended this to handle region-referenced longitudinal EEG data. Xie & Lawniczak demonstrated FDA's value for spectral analysis of epileptic EEG signals. These studies show how FDA methods can effectively handle the high dimensional, complex temporal dependence structure inherent in brain activity data [1].

Data measuring brain activity data are typically high dimensional, complex, and exhibit temporal dependence structure [1], and functional analysis methods offer effective and novel avenues for statistical analysis of neuroscientific data. Developing functional analysis frameworks for neuroscientific data offers the potential for novel insights into complex patterns of variation in functional time series, with less reliance on canonical summary features and the potential for novel insights into influential sources of variation among individuals which fall outside of the scope of traditional multivariate analyses. Our goal in this work is to develop and test a novel methodological framework combining different FDA approaches, demonstrating utility for applied researchers in neuroscience and other fields to analyse functional data and gain novel insights across multiple nested levels of analysis.

When applying FDA to time series data like EEG recordings, the data must first be transformed into a functional form and appropriately smoothed. In this work, we convert EEG time series data into power spectral density (PSD) functions through frequency decomposition, producing curves that represent the distribution of signal power across frequencies at each time point. These PSDs must then be smoothed to create functional data objects - while various approaches like B-spline smoothing can be used for this purpose [28], we employ the 'fitting oscillations and one over f' (FOOOF) algorithm [16] which is specifically designed to separate periodic and aperiodic components in neural power spectra. This transformation assumes that the underlying signal can be meaningfully decomposed into frequency components, and that these components vary smoothly over the frequency domain. An important constraint of our approach is that we analyse data from a single EEG channel, which means we cannot capture spatio-temporal dynamics or conduct analyses of functional connectivity between brain regions. While multi-channel FDA approaches exist [12], we focus on single-channel analysis to maximise clinical utility and interpretability while demonstrating the core principles of our nested functional analysis framework.

FDA encompasses a set of methods to characterise the key modes of variation and identify influential characteristics over the functional domain of observed curves or trajectories. Functional data are typically considered as finite, high dimensional realisations of underlying smooth functions [1,2]. Relative to traditional multivariate approaches, FDA tools are better able to handle the very high dimensional nature of functional data, and offer richer insights on functional data compared to multivariate analyses which are more limited by the 'curse of dimensionality' [2,17]. For data observed over time, the dimensionality is the product of the number of features and the number of time points [10], resulting in issues with the number of dimensions being large relative to the number of observations. To manage challenges around high dimensionality, traditional multivariate statistical methods for analysis of functional neuroscience data observed over time would typically require selecting a small number

of *a priori* features of interest to summarise observed curves, and/or averaging over the temporal dimension. Functional data analysis offers statistical approaches that are more robust to the 'curse of dimensionality' compared to classical multivariate statistics, using tools such as smoothing, regularisation and dimensionality reduction to enable novel insights into influential characteristics across the whole functional domain of interest without restricting focus to *a priori* features of interest [3].

While various approaches exist for analysing functional time series (FTS) data, including direct dimension reduction methods like Functional Singular Spectrum Analysis (FSSA) [18], our goal is to develop a framework that preserves interpretability across multiple levels of analysis while capturing both temporal dynamics and functional characteristics. Recent advances in FTS methodology have provided sophisticated tools for dimensionality reduction and forecasting in both univariate and multivariate contexts [19]. However, for neuroscientific applications, maintaining clear interpretability of outputs at each analytical level is crucial for clinical relevance and practical utility.

One method for functional analysis that offers insights into temporal dynamics within time series of functional data observations is the functional hidden Markov model (FHMM) [22]. Considering the dynamics of a system observed longitudinally, latent state models such as Hidden Markov Models (HMM) [23] can offer useful insights into the latent states that it shifts between over time. HMMs have been extended for use with high dimensional and functional data, which enables analysing curves (functional data) as realisations from unobserved latent states [22].

Recent work has demonstrated the value of HMM approaches specifically for analysing brain state dynamics in neurodevelopmental conditions. Kember et al. applied HMM to resting-state EEG data from the Healthy Brain Network dataset to study network properties in ADHD, identifying distinct electrophysiological states characterised by oscillatory power patterns and finding that dwelling in states with high alpha/beta power supported better response control [20]. Similarly, Shappell et al. used Hidden semi-Markov Models to show that children with ADHD spend less time in anticorrelated network states and more time in hyperconnected states compared to typically developing children [21]. These studies highlight how latent state modeling can reveal important differences in brain dynamics associated with neurodevelopmental conditions.

Another core method for FDA is functional principal component analysis (FPCA). FPCA is able to summarise infinite-dimensional functional data into a finite set of uncorrelated random variables which represent variation over the whole functional domain in a parsimonious way, allowing for the identification of dominant modes of functional variation [2,3]. Functional principal components are similar to traditional principal components which maximise variance explained in vector space for standard PCA, where functional principal components, comprised of eigenfunctions and corresponding eigenvalues or scores, maximise functional variance explained in $L^2$ space [2]. As with traditional PCA, typically a small finite number $C$ of functional principal component and their associated scores are retained for further analysis, which explain a majority of functional variation in the data [3]. Dimensionality reduction with FPCA is a key method for functional analysis to provide insight into dominant attributes of functional variation, and to enable further analysis based on the reduced set of variables which capture much of the information in the original data.

Applying these approaches to neuroscientific data, FHMM outputs can provide rich insight into temporal and frequency characteristics of functional time series data measuring brain activity across individuals. However, it is often of interest to understand in more detail how individuals differ in their functional characteristics when they are allocated to matched states. For instance in the context of sleep studies, a common goal would be to first identify periods

in which individuals are allocated to a common sleep state (e.g. rapid eye movement sleep), before analysing and comparing detailed attributes across individuals within that matched sleep stage type [24,25]. Using Viterbi estimated states from a FHMM model, by selecting subsets of time series data during which individuals are occupying the same latent state, we are able to compare 'like with like' and gain a more detailed understanding of how individuals are similar or different within matched latent states.

In other work developing FDA methods for neuroscientific data, incorporating functional domain information in the same modelling framework as temporal dynamics often comes at the cost of substantially increased model complexity and decreased interpretability. There are several examples in the literature of modelling approaches that simultaneously model brain activity in the frequency, temporal and even spatial domains [1,12,26]. Instead by modelling temporal dynamics of PSDs separately using FHMM and nesting frequency analysis within latent states using FPCA, we are able to generate insights into frequency and temporal dynamics in a way that better preserves interpretability for clinical relevance, while also offering insights into the connections between these levels of analysis.

Building on previous work to analyse characteristics of functional data observed for multiple individuals over time, in this paper we introduce a methodological framework called *flawless* (**F**unctiona**L** **A**nalysis **W**ithin **L**at**E**nt **S**tate**S**), a nested FDA framework for analysis of time-varying functional data that incorporates latent state structure, temporal dynamics and functional characteristics in the frequency domain. By integrating latent state modelling of functional data using FHMM and FPCA of functional characteristics stratified by allocation to matched latent states, this method offers more detailed insights and richer comparisons beyond those available through its component methods. FHMM outputs provide insight into the attributes of distinct latent states that individuals occupy, and the temporal dynamics of their movement between states including the number of states they visit and the number of transitions between them. Subsetting the data based on allocation to latent states, FPCA models for each state provide insight into the dominant characteristics of functional variation in the frequency domain that differentiate individuals within each latent state. While the present analysis has been developed for resting state brain activity data in young people, this framework can readily be applied to other instances of functional data observed over time.

We use *flawless* analysis to understand characteristics of resting state brain activity in early adolescents, using EEG data from the Child Mind Institute's Health Brain Network study (HBN) [27]. HBN is a cross-sectional study covering a large number of data types relating to brain activity, cognitive, physical, and mental health in young people from the New York area. The aim of HBN is to create a large-scale biobank of data to facilitate the discovery of biomarkers and the exploration of prevalent illness phenotypes linked to psychopathology and cognitive function. Using this distinctive data source, we demonstrate the value of *flawless* analysis for generating unique and novel insights regarding characteristics of resting state brain activity in young people, and demonstrate substantial associations between functional analysis outputs and measures of psychopathology and cognitive function.

For resting state brain activity in the frequency domain measured over time, analysis of latent states and trajectories from FHMM and dominant modes of functional variation from FPCA provides valuable insights. Alternative approaches, including traditional multivariate analyses that cannot accomodate functional data, and complex functional methods that simultaneously model multiple functional domains, have different analytical goals and produce outputs which are not readily comparable to *flawless* analysis. Given the limited ability in this context to directly compare quantitative performance or model fit metrics against related methods, instead we focus in this work on addressing qualitative differences and highlighting unique features that are available through the nested approach of the *flawless* framework.

We address the following applied research questions regarding the dynamics of resting state brain activity in young people, and their associations with psychopathology and cognitive function:

1. Is functional analysis within latent states an effective approach to implement for characterising resting state brain activity measured by EEG?
2. What are the characteristics of functional latent states that young people occupy during eyes closed, resting state brain activity? How do measures of psychopathology and cognitive function vary between groups of individuals who spend a majority of time in each one of these states?
3. What are the temporal dynamics of movement between functional latent states? How many states do individuals visit, what proportion of time do they spend in each state, and how frequently do they move between them?
4. Examining subsets of functional data by allocation to latent states, what frequency characteristics differentiate brain activity among individuals in each state?
5. Considering outputs of these functional analyses in terms of latent states, temporal dynamics and frequency content within latent states, how are these traits associated with measures of psychopathology and cognitive function?

The rest of the paper is organised as follows. In the Methods we provide an overview of the methodological pipeline for *flawless* analysis, providing information on the background and implementation of the component methods, and provide details on neuroscientific and health outcome data collected in the HBN study. In the Results we present findings from the different stages of our analysis of resting state EEG characteristics in adolescents, including functional latent states and temporal dynamics of state occupancy from the FHMM, health measures and frequency characteristics across groups of individuals with 80% or more of their time allocated to one latent state. We then present results of Bayesian regression models demonstrating associations between *flawless* analysis outputs and health measures relating to psychopathology and cognitive function. In the Discussion we discuss the benefits and implications of this method and our findings, and consider limitations and future directions for this work.

## Methods

In this section we provide an overview of the methodological steps involved in the *flawless* analysis framework. We then introduce the study protocols and data collection details for neuroscientific and health measures assessed in the HBN study, before describing pre-processing, frequency decomposition and extraction of smoothed representations of periodic content from EEG data. Background and implementation details are provided for FHMMs, functional principal component analysis, and Bayesian regression models. The Healthy Brain Network study was approved by the Chesapeake Institutional Review Board. Prior to conducting the research, written informed consent is obtained from participants ages 18 or older. For participants younger than 18, written consent is obtained from their legal guardians and written assent obtained from the participant.

### *Flawless* overview

We begin with a high level overview of the steps involved in *flawless* analysis. The intention is to provide a 'road map' for the reader, making the detailed explanations of each individual step easier to understand in the broader context of this framework. Details on notation

are provided in the subsequent sections, and a reference Table 5 describing key notation is provided at the end of this manuscript. The method implemented in the *flawless* analysis framework consists of the following steps:

1. Taking a time series $\mathbf{X}_{pk}$ of functional data observations indexed over time $k$ from one or more individuals/units $p$, perform some initial smoothing of the data over the functional domain. In general applications, an approach such as B-spline smoothing may be appropriate for preparation of data prior to fitting the FHMM [28]. In the present work we generate smoothed representations of the periodic content in the frequency domain for EEG power spectral densities using the FOOOF algorithm [16].

2. Fit a FHMM to the time series of smoothed functional data, finding a set of $N$ functional latent states across individuals to characterise the different states occupied over time [22]. An initialisation strategy based on multiple subsampled models may be used to identify the number of latent states $N$ and improve model stability through selection of well-separated initial centroids, described in the Methods and the Supplementary Materials.

3. Use the Viterbi algorithm to calculate the maximum a posteriori estimated series of latent states $\mathcal{V}$ associated with the series of observed functional data observations [29]. Based on this estimated series of states, calculate summary statistics for each individual including the number of states occupied, percentage of time spent in each state, and number of transitions between states.

4. Create subsets of the input data $\mathbf{X}_{pk}$ based on allocation to each state $s_i$, $i = 1, ..., N$ in the vector of state allocations $\mathcal{V}$ estimated for each individual in (3), $\mathbf{X}_i = \mathbf{X} \in s_i$.

5. For each subset $\mathbf{X}_i$ of functional data observations allocated to each latent state, apply functional principal component analysis. For each FPCA model, retain an appropriate number of functional principal components $C$ based on inspection of scree plots and cumulative percentage of variance explained.

6. Based on the within-state functional principal components from (5) and the FHMM outputs from (3), these outputs may be used for subsequent analyses such as regression or clustering to examine relationships between functional characteristics and associated outcome variables.

For clarity and ease of interpretation, this is also presented in Algorithm 1 as pseudocode. Fig 1 also presents a methods diagram indicating an overview of the components of the *flawless* analysis framework and their application to functional data calculated from resting state EEG recordings in the present application.

## The Child Mind Institute's Healthy Brain Network study

The Child Mind Institute's Healthy Brain Network study is a large scale initiative collecting data on a variety of measures relating to brain activity, physical health, mental health and cognitive development in young people aged 5 - 21 years in New York City and surrounding areas [27]. The study uses a community referral-based recruitment model to encourage families concerned about psychiatric symptoms in their child to participate. The goal for HBN is

**Algorithm 1.** *flawless* (FunctionaL Analysis Within LatEnt StateS)

```
   Notation:
   X_pk : Time series of functional data observations
   p : individuals/units
   k : time points
   t : functional domain
   a.b : Access field/attribute b of object a (object-oriented
   notation)
1: procedure Flawless(X_pk)
2:    V : Vector of state allocations
3:    FPC_i : Functional principal components for each state i
4:    Stats_p : Summary statistics for each individual p
5:    // Smooth functional data over domain t
6:    X_smooth ← SmoothData(X_pk)          ▷ e.g., B-spline smoothing or
   FOOOF for EEG PSDs
7:    // Fit FHMM to identify latent states
8:    N ← DetermineStateNumber(X_smooth)              ▷ via subsampling
   strategy to choose initial centroids
9:    States ← FitFHMM(X_smooth, N)
10:   // Calculate state allocations and statistics
11:   V ← ViterbiAlgorithm(States, X_smooth)
12:   for each p do
13:      Stats_p.n_states ← CountUniqueStates(V_p)
14:      Stats_p.state_proportions ← CalculateStateTimes(V_p)
15:      Stats_p.n_transitions ← CountTransitions(V_p)
16:   end for
17:   // Create state-specific data subsets
18:   for i = 1 to N do
19:      X_i ← SubsetDataByState(X_smooth, V, state = i)
20:   end for
21:   // Perform FPCA within each state
22:   for i = 1 to N do
23:      C ← DetermineComponentNumber(X_i)         ▷ via scree plots
24:      FPC_i ← FitFPCA(X_i, C)
25:   end for
      return V, FPC, Stats
26: end procedure
```

to generate a large-scale biobank of data for biomarker discovery and investigations of commonly occurring illness phenotypes relating to psychopathology and cognitive function. Relative to a population sample, the strategy of recruiting on the basis of perceived clinical concern means that the HBN sample includes a high proportion of individuals with elevated psychopathology and/or cognitive difficulties. As the present study focuses on brain activity, psychopathology and cognitive function in the period of early adolescence, we use data from participants aged 9 to 15 years in the HBN biobank. A number of other studies have used this age range for early adolescence [30–32], and we used this range to maximise the number of participants included, while confining our focus to this developmental period of interest.

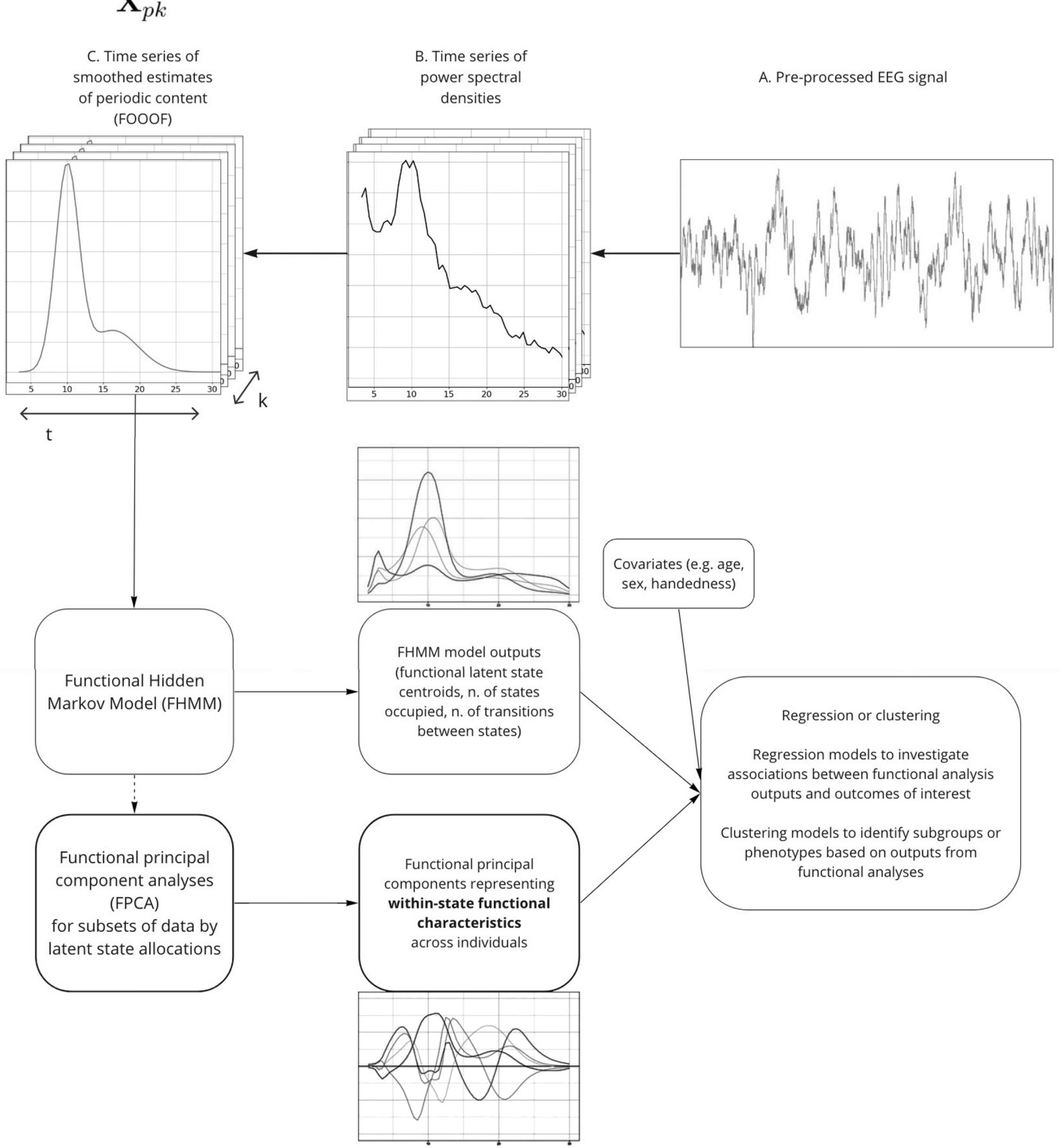

**Fig 1. Flawless analysis framework overview, with application to EEG power spectral densities as functional data observed over time.** $X_{pk}$ indicates a functional data array with $p$ = individuals, $k$ = time, and $t$ = functional domain. FOOOF = fitting oscillations and one over f algorithm - see Methods for details.

Data were downloaded from the HBN portal on July 4, 2022. Data used for this study were contained in Release Numbers 1.1 - 10 of HBN, with Release Dates between January 31, 2018 and April 13, 2022. The authors did not have access to information that could identify individual participants during or after data collection.

**EEG data acquisition.** For this paper we use resting state, eyes closed EEG data, recorded at the vertex electrode Cz. We chose to focus on single electrode, eyes closed, resting state data for the sake of generalisability and clinical utility. Compared to task-based EEG data, resting state EEG data offers more opportunities for clinical applications as it is easier to replicate data collection across different laboratory and clinical settings without requiring specific software and equipment for task-based paradigms [33,34]. It is also simpler for data cleaning and preparation, with potential for straightforward implementation on a wider scale. We focused specifically on the Cz electrode, which is often used as a reference electrode due to its central location and typically high-quality signal [76]. Cz was used as the reference electrode in the HBN EEG acquisition protocol. This placement provides a good balance between signal quality and reduced susceptibility to artifacts, and the consistent placement of Cz could support implementation across various clinical and research settings [77]. These factors enhance the potential clinical utility of our findings, as data-driven phenotypes uncovered from these measures will be straightforward to measure and implement across a variety of scenarios for EEG acquisition in clinical and research settings.

During resting state recording sessions, participants viewed a fixation cross in the center of a computer screen. Throughout the paradigm, participants were instructed to open or close their eyes at various points, alternating between 40 second periods of eyes closed and 20 second periods of eyes open for a total of 5 minutes (300 seconds total, with 200 seconds of eyes closed recording). The paradigm was designed to measure endogenous brain activity during rest [27]. High-density EEG data were recorded in a sound-shielded room at a sampling rate of 500 Hz with a bandpass of 0.1 to 100 Hz, using a 128-channel EEG geodesic hydrocel system by Electrical Geodesics Inc. Full details of EEG data acquisition including cap fitting, impedance checking and preparation are available in the HBN data descriptor [27].

**Psychopathology and cognitive function measures.** A wide variety of measures regarding physical, mental and cognitive health have been included at various time points in the evolving HBN study protocol. We selected four measures of psychopathology and four measures of cognitive function, motivated by choosing measures which captured a broad range of outcomes in each domain and were available for a majority of participants since early timepoints in the HBN study.

The psychopathology measures we selected were: Mood and Feelings Questionnaire, Self Report (MFQ SR) which measures depression symptoms in children and young people [35]; Screen for Child Anxiety Related Emotional Disorders, Self Report (SCARED SR), which measures anxiety symptoms in young people [36]; and the internalising and externalising scales from the Youth Self Report inventory (YSR Int and YSR Ext), which measure broad dimensions of internalising and externalising psychopathology [37,38]. The four measures of cognitive function were from the National Institutes of Health (NIH) Toolbox Cognition Battery [39]. These measures included the Card Sorting task (NIH Card) measuring executive function, the Flanker task (NIH Flanker) measuring executive function and attention, the List Sorting task (NIH List) measuring working memory, and the Pattern Comparison task (NIH Pattern) measuring processing speed.

## EEG processing, frequency decomposition and smoothing

EEG data used in this paper were pre-processed internally by HBN investigators, using steps including identification and replacement of electrodes with poor data quality, high pass filtering at 0.1 Hz, notch filtering at 59-61 Hz to remove background electrical line noise, and removal of eye movement artifacts. Full details of EEG cleaning and pre-processing steps are available in a separate publication describing methodology for the Multimodal Resource for Studying Information Processing in the Developing Brain (MIPDB), another study run by the Child Mind Institute [40]. Following standardised EEG processing methodology used in MIPDB, EEG data were filtered between 1.5 and 30 Hz prior to frequency decomposition, covering the canonical frequency bands between delta (1.5 - 4 Hz) and beta (14 - 30 Hz).

To study frequency content of EEG data we used multitaper analysis, a frequency decomposition method that is suitable for non-stationary signals and offers good frequency specificity [41]. This is a popular technique for frequency analysis that has been used widely in recent EEG research literature [42,43], and offers an improved signal-to-noise ratio for detecting rhythmic activity in a signal relative to other standard frequency decomposition methods [44,45]. Multitaper analysis was conducted using the Chronux toolbox in MATLAB [46]. This method enabled us to calculate power spectral densities with good frequency resolution and minimal 'bleeding' of power across adjacent frequency bands [47]. This analysis used a 2 second time segment (W) and a 2Hz frequency bandwidth (W), resulting in a total of 3 tapers (2TW - 1). These parameters were chosen to provide adequate frequency resolution, giving a detailed characterisation of the frequency distribution in each individual's power spectral density [43].

Multitaper analysis initially generated a time series of 300 power spectral densities for each individual, representing the frequency content of their EEG signal over the 300 seconds of resting state recording time, recorded on the vertex electrode Cz as described above. As we were interested in activity recorded when participants had their eyes closed, we trimmed out each eyes closed segment with a 2 second leading and trailing buffer, resulting in 180 seconds total of eyes closed PSDs per individual across 5 segments each with 36 PSDs.

Finally we applied the FOOOF algorithm ('fitting oscillations and one over f') in order to identify periodic components of EEG activity, removing aperiodic power and generating smoothed estimates of the periodic content in PSDs which could be used as inputs for the FHMM. The FOOOF library in Python (version 1.0.0) was used to parameterise neural power spectra. Settings for the algorithm were set as: peak width limits: 1 - 12 Hz; max number of peaks: no limit; minimum peak height: 0.05; peak threshold: 2.0; and aperiodic mode: fixed, without a knee. Power spectra were parameterised across the frequency range 1.5 to 30 Hz. Performance was assessed based on visual comparison of FOOOF model fits against original PSDs, and on metrics for model error and goodness of fit. The outputs retained from the FOOOF algorithm were the peak fit objects, representing the peaks in each PSD with aperiodic power subtracted, reconstructed as a sum of Gaussians fit to the centre frequency, amplitude and width of each detected peak. This produced smoothed representations of the periodic oscillatory content in each PSD, which were suitable as inputs for subsequent functional analyses.

## Functional hidden Markov model

**Background.** Hidden Markov Models (HMMs) are a type of statistical model that use a Markov process with hidden states to make probabilistic models of linear sequence labeling problems, and can be used to characterise the modes or states that a system occupies and moves between over time [48]. These models are designed to handle time series data by using

emitted symbols that are observable realisations from latent states, and random transitions from one latent state to another that remain unobserved. The memory-less property of the Markov chain, where the transition from one state to another depends only on the present state, is a key concept in the HMM framework.

A Hidden Markov Model is a bivariate process $\{(Q_k, X_k)\}_{k \geq 1}$ defined on a given probability space $(\Omega, \mathcal{F}, \mathbb{P})$ such that:

- $\{Q_k\}_{k \geq 1}$ is a Markov chain with a discrete and finite state space $\{s_1, \ldots, s_N\}$, with $N \geq 1$, transition matrix $\mathbf{A} = \{a_{ij}\} = \mathbb{P}(Q_k = s_j \mid Q_{k-1} = s_i)$ and initial distribution $\nu$, where $\nu_i = \mathbb{P}(Q_1 = s_i)$;
- For each time $k$, the observation $X_k$ is a $d$-dimensional random array. In particular, given the state process $\{Q_k\}_{k \geq 1}$, $X_k$ is a sequence of conditionally independent random arrays (vectors or matrices, depending on the type of data) [22,48].

In the general case, the objective function for a HMM can be written as:

$$
\log(\mathcal{L}(\lambda \mid \mathbf{x})) = \sum_{i=1}^{N} \gamma_1(i) \log \nu_i + \sum_{i=1}^{N} \sum_{j=1}^{N} \left( \sum_{k=1}^{K-1} \xi_k(i,j) \right) \log a_{ij}
$$
$$
+ \sum_{i=1}^{N} \sum_{k=1}^{K} \gamma_k(i) \log b_i(\mathbf{x}_k; \theta_i).
$$

(1)

where $\gamma_k(i) = \mathbb{P}(Q_k = s_i \mid \mathbf{X}_1 = \mathbf{x}_1, \ldots, \mathbf{X}_k = \mathbf{x}_k, \lambda)$ is the probability of being in the state $s_i$ at time $k$, $\xi_k(i,j) = \mathbb{P}(Q_k = s_i, Q_{k+1} = s_j \mid \mathbf{X}_1 = \mathbf{x}_1, \ldots, \mathbf{X}_k = \mathbf{x}_k, \lambda)$ is the probability of being in state $s_i$ at time $k$ and state $s_j$ at time $k+1$, given the model and the observations, and $b_i(\mathbf{x}_k; \theta_i)$ is the emission function of $\mathbf{X}_k$ conditionally on the event $\{Q_k = s_i\}$ for any $i = 1, \ldots, N$. The Baum-Welch algorithm can then be used to compute the value in (1) and iteratively perform expectation maximisation and the 'forward-backward' procedure to calculate the objective function and the best estimates for model parameters [49,50].

In a standard HMM, the emission function is a probability distribution that describes the likelihood of observing a particular output symbol (or emission) given the current hidden state. It describes the relationship between the unobservable internal state of the system and the observable data, enabling inference about the underlying hidden states from the observed data. For a functional observation $\mathbf{x}_k$, the emission function of $\mathbf{x}_k$ conditionally on the event $\{Q_k = s_i\}$ is represented as $b_{\mathbf{X}_k \mid Q_k = s_i}(\mathbf{x}_k; \mu_i)$, for any $i = 1, \ldots, N$, where $\mu_i$ is a functional parameter representing the mean of the curves emitted by state $s_i$. In a FHMM, the emission functions $b_{\mathbf{X}_k \mid Q_k = s_i}(\mathbf{x}_k; \mu_i), i = 1, \ldots, N$ are constructed based on distances between curves. Specifically, this method assumes that for each state $s_i$, the emission function can be written as

$$
b_{\mathbf{X}_k \mid Q_k = s_i}(\mathbf{x}_k; \mu_i) = h(d(\mathbf{x}_k, \mu_i)), \quad i = 1, \ldots, N
$$

(2)

where $h : \mathbb{R} \to \mathbb{R}$ is a function that transforms the distance into a similarity measure. In particular, the implementation by Martino et al. (2020) uses the function $h(y) = 1/y^2$ and the $L^2$ distance for $d$. For a full discussion of the development and methodology of the functional HMM, please refer to [22].

Following identification of functional latent states from a FHMM, a number of insights may be gained from interpretation of the fitted FHMM model characteristics and estimation of the most likely sequence of latent states. One common approach is to use the Viterbi algorithm [51] to calculate the most likely sequence of states to have generated the observed

data. From the generated sequence of most likely latent states, further analysis may identify attributes including dominant states which were visited by a higher number of individuals, and inter-individual comparisons including the time spent in each state and the number of states visited.

**Implementation.**   For this analysis, we implement a FHMM using the *hmmhdd* R package, version 1.0 [52]. As described above, the functional data time series input to the model are peak fit outputs from the FOOOF algorithm, representing a smoothed estimate of the periodic oscillatory content in each power spectral density.

To manage instability issues arising from initialisation of functional latent state centroids based on a functional $k$-means algorithm, we used a subsampling strategy described below to identify an appropriate number of stable and robust centroids present in the data to initialise the model.

**Initialisation strategy for FHMM latent state centroids.**   During the development of this work, we identified instability issues in the performance of FHMMs, as implemented in the R package *hmmhdd* [22,52]. Through testing, we discovered an issue where the FHMM algorithm tends to find multiple latent states with identical centroids. Our understanding is that this instability likely occurs due to the use of a functional $k$-means algorithm to initialise the latent state centroids for the FHMM [53,54]. $k$-means with random initialisation has known performance issues where the clustering results can have a high degree of instability and are very sensitive to the initial conditions [55]. This is particularly challenging for clustering functional data, as they are typically very high dimensional, so the $k$-means algorithm with random initialisation appears to have a tendency for multiple clusters to collapse towards the same local maximum in the high-dimensional space, often resulting in near-identical centroids across multiple functional latent states.

In response to this issue, we developed a subsampling based initialisation strategy to improve the stability and performance of the FHMM algorithm. To do this, we fit 10 FHMM models based on 80% randomly subsampled sets of the full data in order to identify stable centroids. 80% was used as a subsampling proportion based on common practice in the literature [56,57], removing a small proportion (20%) of the data at random in order to identify stable and robust results across subsamples. For this application, based on testing with HBN data these subsampled models were each set to identify 8 latent states, allowing for redundant overlapping states to occur while enabling identification of stable consistent states. As a result of this instability in FHMM performance, model selection based on comparing information criteria between candidate models was not a feasible strategy to choose the number of latent states $N$. Instead, taking all centroids identified across the subsampled models, we plotted them together to allow visual identification of consistent and well-separated state centroids which appear across models. These heuristically grouped centroids across subsampled models were then averaged and used to generate a set of $N$ initial centroids for the FHMM, to prevent generation of redundant states with identical centroids. More details on this process are provided in S1 and S2 Figs.

## Functional principal component analysis

**Background.**   Functional principal component analysis is an extension of principal component analysis for dimensionality reduction, and it can be used to analyse dominant modes of variation across the functional domain for data sets consisting of functions or curves. FPCA enables representation of the infinite-dimensional functional data as a finite-dimensional vector of random scores, which can subsequently be modeled using the tools of multivariate data analysis. This method is based on an expansion of the underlying random

trajectories in a functional basis consisting of the eigenfunctions of the covariance operator of the process [2]. The resulting FPCs or scores capture the dominant modes of variation in the data and can be truncated to a finite vector, achieving the goal of dimensionality reduction. Like a traditional PCA explaining maximum variance in principal components consisting of eigenvectors and eigenvalues, FPCA decomposes functional data into eigenfunctions and eigenvalues, capturing variation in curves across the whole functional domain.

For a set of functional data $X_p(t)$ where $t$ is the functional domain, the FPCA expension is as follows:

$$X_p(t) = \mu(t) + \sum_{c=1}^{\infty} A_{pc}\phi_c(t), \tag{3}$$

where $\mu(t)$ is the mean function of $X_p(t)$, $\phi_c$ are the orthogonal eigenfunctions, and $A_{pc} = \int_P \left( X_p(t) - \mu(t) \right) \phi_c(t)\mathrm{d}t$ are the functional principal components of $X_p$. This expansion in (2) enables dimensionality reduction as the first $C$ terms that explain a substantial amount of overall functional variance provide a good approximation to the infinite sum, so that the information contained in $X_p$ is largely contained in the $C$-dimensional vector of eigenvalues $\mathbf{A}_p = \left( A_{p1}, \dots, A_{pC} \right)$ and the approximated processes

$$X_{pC}(t) = \mu(t) + \sum_{c=1}^{C} A_{pc}\phi_c(t). \tag{4}$$

Based on assessment of scree plots and the cumulative percentage of functional variance explained, a small finite number $C$ of functional principal components can be retained which explain the majority of variation and represent a parsimonious dimension-reduced summary of the original functional data.

**Implementation.** FPCA models were fit to subsets of the time series of PSDs in this dataset, stratified by allocation to functional latent states generated from the Viterbi algorithm. This was implemented in the R package *fda*, version 6.0.5 [58]. For each subset of functional data by state, FPCA was performed using penalised smoothing to fit a series of B-spline basis functions to the FOOOF PSD curves for each individual and each time point [7]. To select functional PCs to retain, we assessed scree plots by looking for the 'elbow' point where the rate of decrease in explained variance begins to level off. On inspection of these plots, we looked for the number of components that cumulatively explained a substantial proportion (> 70%) of the total variance while balancing parsimony.

## Bayesian regression models for mental health and cognitive function

To examine patterns of association between outputs from functional analyses and health measures relating to psychopathology and cognitive function, we implemented Bayesian regression models using the *brms* package in R, version 2.15.0 [59,60]. We fit separate multivariate response regression models for four subsets of participants, based on the dominant latent state in which participants spent the most time. The output variables for these models were scores on the four measures of psychopathology and four measures of cognitive function, as described above. Independent variables in these regression models included the number of functional latent states visited by each individual, the number of transitions between states over the resting state recording period, the percentage of time spent in the dominant state, and scores on the first five functional principal components for the dominant state. Other independent variables included as control covariates included age, sex, and handedness (measured by the Edinburgh Handedness Questionnaire) [61]. Default settings in *brms* were used

for 'flat' uniform prior distributions on regression coefficients, covering the expected range of the parameter values.

## Canonical EEG frequency bands

As we have indicated in the Introduction, canonical EEG frequency bands do not correspond to functionally distinct categories of brain activity and have substantial flaws for interpretation of differences in oscillatory content across individuals, especially in childhood and adolescence when oscillatory content in brain activity is rapidly changing [16,62]. For ease of description below, we refer to the following labels for frequency ranges: delta (1.5-4 Hz), theta (4-7 Hz), alpha (7-14 Hz), beta-1 (14-22 Hz), and beta-2 (22-30 Hz). There is substantive variation in the definitions of these bandwidth ranges in the literature [34,63,64]. We have based these specific bandwidth labels on evidence that the alpha oscillatory rhythm appears across a wider frequency range than typically used (such as 8-12 Hz) [64], and we used the beta-1 and beta-2 bands implemented by Rogala et al. [34]. However, it is important to note that we are not calculating power within these bands for our principal analyses. The functional analysis methods used here are attuned to the shape of each PSD curve across the whole frequency range considered, and these labels are only used as a heuristic label to support understanding and interpretation pf power distribution across this frequency range, due to their widespread use and familiarity in EEG research.

## Results

We present the results of this analysis in three sections: First we present the results of the FHMM including frequency characteristics of centroids for functional latent states, and differences in psychopathology and cognitive function between individuals who spent 80% or more of their time in each state. The second section presents results of FPCA analyses stratified by latent states, identifying the dominant modes of functional variation that distinguish frequency content among individuals in matched latent states. The third section presents results of Bayesian regression models investigating associations between outputs from FHMM and FPCA models with outcome variables relating to psychopathology and cognitive function.

In this work we used pre-processed resting state EEG data, which were available for n = 503 early adolescents between the ages of 9 and 15 years (M = 11.5, SD = 1.7) in the HBN study. Descriptive statistics for demographics, psychopathology and cognitive function are presented in Table 1. As the HBN study protocol has evolved over time, some measures introduced later in the study (including the Youth Self Report scale) are available for fewer participants. As noted in the Methods, the HBN study recruits participants using a targeted recruitment strategy for children with mental health and cognitive difficulties, and so this dataset exhibits higher levels of psychopathology and lower cognitive function than would be expected in a population sample. 'Fitting oscillations and one over f' (FOOOF) performance metrics indicated good performance for models fit to estimate aperiodic content in PSDs, with a mean $R^2$ of 0.964, and an average of 3.6 peaks identified per PSD [16].

## Functional hidden Markov model

Based on centroids that appeared consistent and well-separated across 10 FHMMs fit using 80% random subsamples of the full data, we identified 4 functional latent states. Further details on FHMM initialisation based on stable centroids across subsampled models are provided in S1 and S2 Figs.

**Table 1. Demographics, cognitive function and psychopathology measures for the overall sample. Last column gives the Mean (SD).**

|  | N Obs. | Overall ($n = 503$) |
|---|---|---|
| **Demographics** | | |
| Female - N (%) | 503 | 165.0 (32.8%) |
| Age | 503 | 11.5 (1.7) |
| EHQ | 491 | 61.6 (48.4) |
| **Cognitive Function** | | |
| NIH Card | 496 | 93.9 (16.9) |
| NIH Flanker | 494 | 87.0 (13.3) |
| NIH List | 489 | 97.9 (15.4) |
| NIH Pattern | 495 | 92.4 (22.3) |
| **Psychopathology** | | |
| MFQ SR | 447 | 13.4 (10.4) |
| SCARED SR | 490 | 23.0 (15.9) |
| YSR Ext | 292 | 51.8 (10.3) |
| YSR Int | 292 | 55.8 (11.5) |

EHQ = Edinburgh Handedness Questionnaire; NIH = National Institutes of Health Toolbox Cognitive function tasks; NIH Card = Card Sorting task measuring executive function; NIH Flanker = Flanker task measuring executive function and attention; NIH List = List sorting task measuring working memory; NIH Pattern = Pattern comparison task measuring processing speed; MFQ SR = Mood and Feelings Questionnaire, Self Report; SCARED SR = Screen for Child Anxiety Related Disorders, Self Report; YSR Ext = Youth Self Report, Externalising Scale; YSR Int = Youth Self Report, Internalising Scale.

**Frequency characteristics of functional latent states.** Fig 2 displays centroids for the 4 functional latent states, labelled by decreasing frequency. State 1 (red) is the most commonly visited state, being the most occupied state for n = 213 individuals, and is characterised by high relative power in the delta range (1.5-4 Hz), very low alpha power (7-14 Hz), and high beta-2 power (22-30 Hz). State 2 (orange) is the dominant state for n = 139 individuals, and has high relative theta power (4-7 Hz), a low frequency, moderate power alpha peak at 9 Hz, and low beta-1 power (14-22 Hz). State 3 (green) is the dominant state for n = 79 individuals, and has low relative theta power, a high frequency, moderate power alpha peak at 11 Hz, and high beta-1 power (14-22 Hz). State 4 (blue) is the dominant state for n = 72 individuals, and has low relative delta and theta power, very high alpha power with a moderate frequency alpha peak at 10 Hz, and low beta-2 power.

**Latent state occupancy patterns.** Fig 3 presents a bar chart of the number of individuals for whom each state is dominant, and histograms of the number of states occupied and the number of transitions between states. State labels have been ordered by decreasing frequency of allocation. A majority of individuals visit only 1 (n = 234; 46.5%) or 2 (n = 205; 40.8%) latent states, and a majority (n = 479; 95.2%) make between 0 and 4 transitions between states over the recording period.

Fig 4 presents a Venn diagram representing the combinations of states that individuals visit during resting state recordings, as captured in the sequence of estimated states generated by the Viterbi algorithm. The most common combinations of states visited were: State 1 only (25.6%); States 1 and 2 (21.5%); State 2 only (9.3%); States 3 and 4 (6.4%); State 3 only (6.0%); and State 4 only (5.6%).

**Comparing psychopathology and cognitive function between dominant states.** Given the variation in the proportion of time which individuals spend in their most common state, it is of interest to contrast relevant health measures between individuals who spend a substantial majority of their time in each state, in order to understand the dominant patterns of

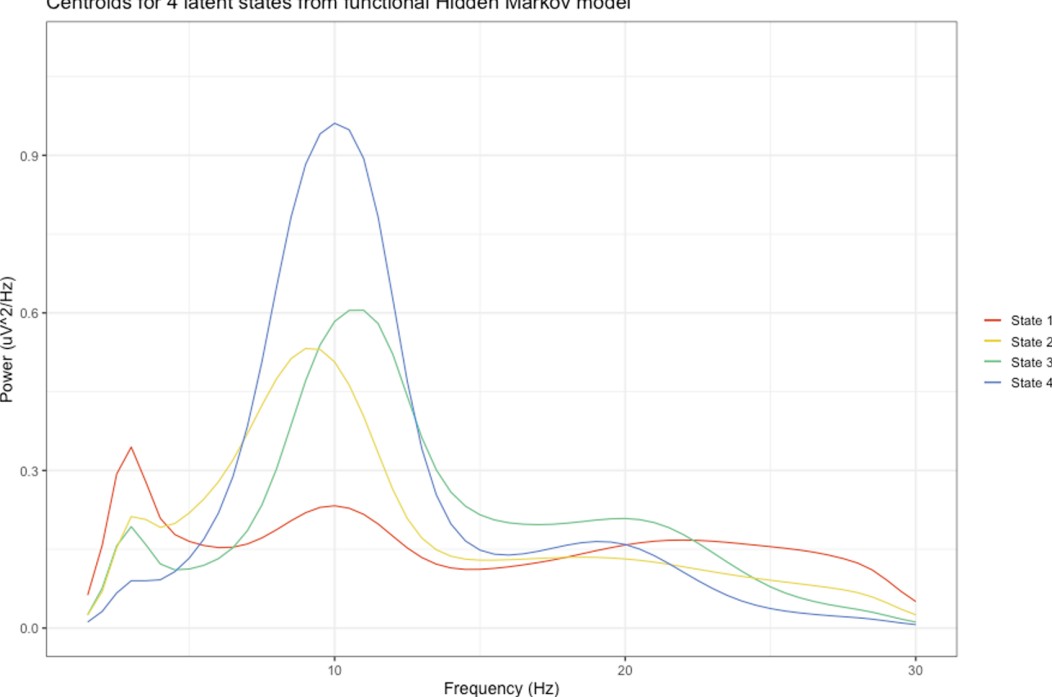

**Fig 2. Centroids for 4 functional latent states from functional hidden Markov model.**

variation in psychopathology and cognitive function between states. This descriptive analysis aims to characterise patterns in psychopathology and cognitive function between individuals who spent substantial time in each state, providing context for subsequent detailed analyses, rather than make inferential claims about statistical significance or broader generalisability of these differences. For Fig 5 and Table 2, data is included for individuals who spent 80% or more of their time in one state during resting state recordings (n = 353; 70.2%). Fig 5 presents bar charts indicating Z-scores (scaled and centered values) for 4 cognitive testing measures from the NIH Toolbox, and 4 measures of psychopathology. These plots are based on data presented in Table 2. Frequentist analyses of variance (ANOVAs) revealed no significant differences in sex or handedness between these groups, but there was a statistically significant difference ($p = 0.001$) in age detected between these groups. Full ANOVA results are available in S1 Table. Differences in age, sex and handedness between groups were accounted for by including these demographic factors as covariates in Bayesian regression models below.

Fig 5 and Table 2 show that individuals who spent 80% or more of their time in State 1 (n = 167) had poorer cognitive function and higher psychopathology than the overall average, based on multiple measures. These included below average scores for the NIH Card sorting and Flanker tasks, indicating poorer executive function and attention. They also had above average scores for the MFQ SR, YSR Externalising and YSR Internalising scales, indicating higher depressive symptoms and psychopathology. This suggests that spending a majority of resting state time in State 1 may be a risk marker for elevated cognitive difficulties and psychological distress.

For State 2 (n = 90), these individuals had improved cognitive scores on the NIH Card and Flanker tasks, as well as the Pattern task which measures processing speed. However, they also

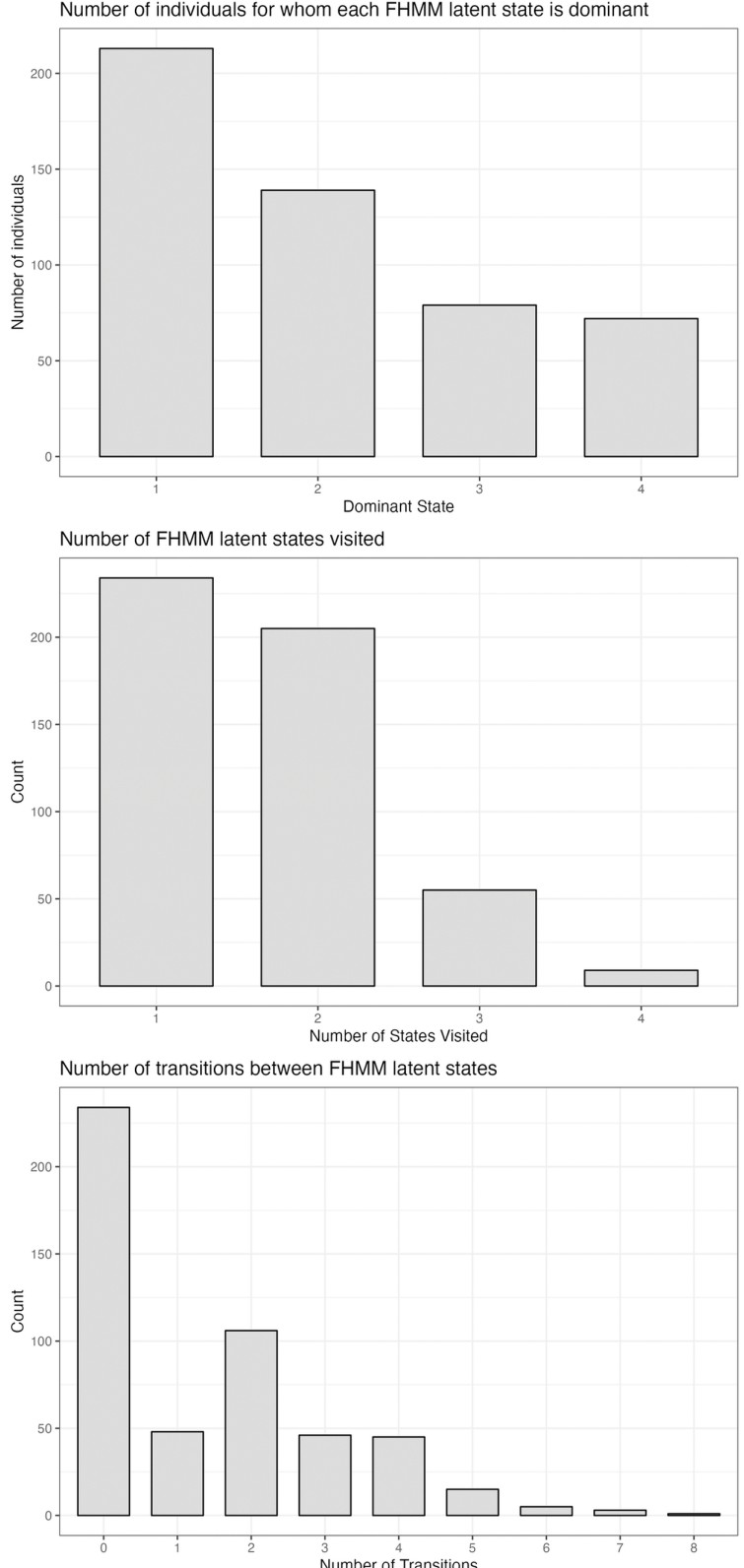

**Fig 3. Bar charts of the number of individuals for whom each state is dominant, the number of states occupied and the number of transitions between states.**

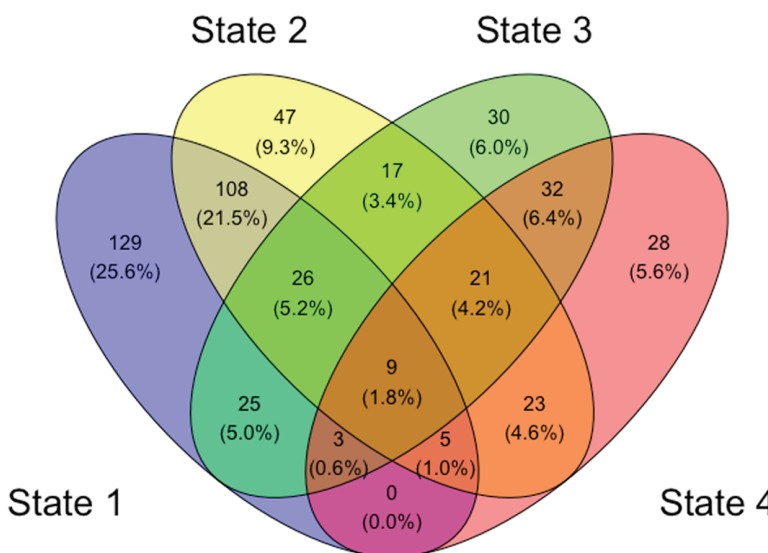

**Fig 4. Venn diagram of FHMM states visited by individuals.** Numbers in cells represent the number of individuals who visited that combination of functional latent states.

had elevated scores for all four psychopathology measures including anxiety symptoms measures with the SCARED SR scale. This suggests that extended time in State 2 may be a risk indicator for increased psychopathology, alongside higher cognitive function.

Individuals spending 80% or more of their time in State 3 (n = 52) exhibited improved cognitive function and lower psychopathology, including higher than average scores on the NIH Card, Flanker and Pattern tasks, and low scores on all four psychopathology measures. This suggests that spending a majority of time in State 3 may be indicative of a beneficial marker for improved mental health and cognition.

State 4 (n = 44) had the smallest number of individuals spending 80% or more of their time, and had a more mixed profile with 3 cognitive function scores above average and 1 below, as well as 3 psychopathology scores above average and 1 below. Given the smaller group size and mixed direction of these effects, the interpretation of health measures associated with this group is less clear.

## Functional principal component analyses within latent states

Based on the latent states identified using the Viterbi estimated state sequence from the FHMM, we split the data into subsets of PSDs allocated to each latent state. For each subset, we ran a separate FPCA model in order to identify the dominant modes of functional variation in the frequency domain that differentiate individuals occupying matched latent states.

For the sake of space, in the body of the text below we present FPCA and regression results for functional latent states 1 and 3. Based on the results above, State 1 is the most commonly occupied, and individuals who spent 80% or more of their time in this state had higher psychopathology and lower cognitive function scores, indicating that this is a state of concern which warrants further detailed investigation. State 3 has a notable pattern of lower psychopathology and higher cognitive function scores relative to the sample average, indicating that time spent in this state may indicate lower risk of psychological distress, and improved

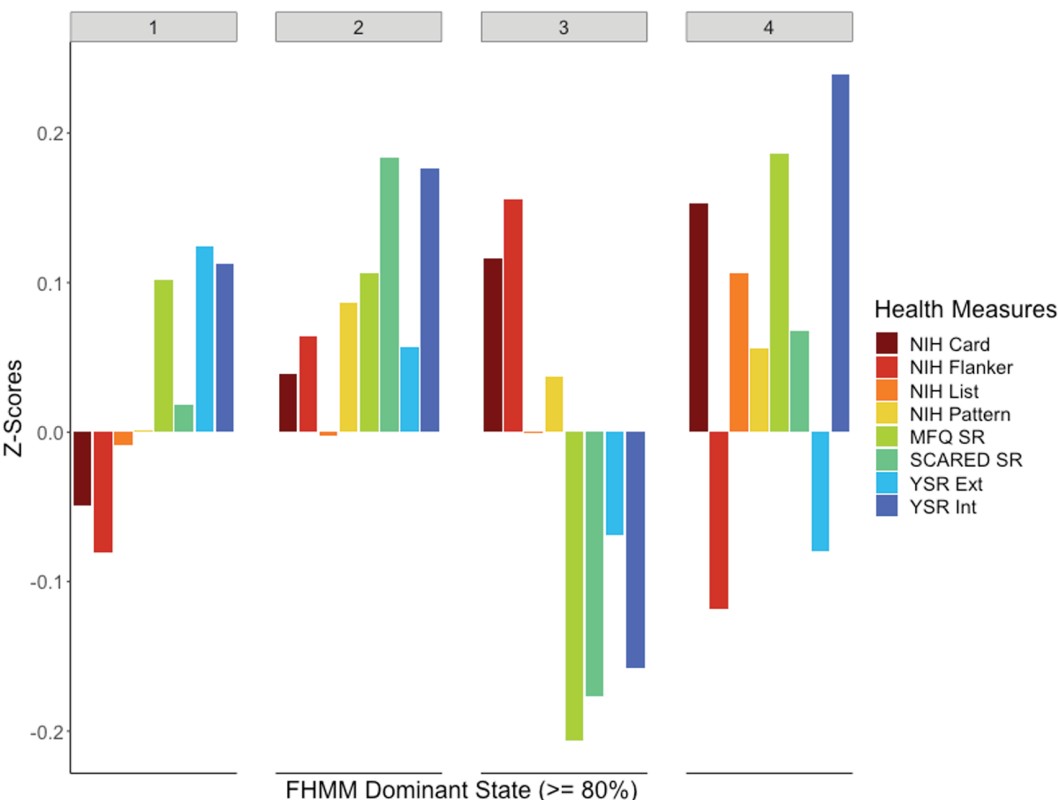

**Fig 5. Bar charts comparing Z-scores (scaled and centered values) for 4 cognitive function measures and 4 psychopathology measures, between individuals who spent 80% or more of their time in each state.** NIH = National Institutes of Health Toolbox Cognitive function tasks; NIH Card = Card Sorting task measuring executive function; NIH Flanker = Flanker task measuring executive function and attention; NIH List = List sorting task measuring working memory; NIH Pattern = Pattern comparison task measuring processing speed; MFQ SR = Mood and Feelings Questionnaire, Self Report; SCARED SR = Screen for Child Anxiety Related Disorders, Self Report; YSR Ext = Youth Self Report, Externalising Scale; YSR Int = Youth Self Report, Internalising Scale.

cognitive health outcomes. Full results for states 2 and 4 are available in the Supplementary Materials.

**State 1 – FPCA.** For State 1, we retained 5 functional principal components which cumulatively explained 72.6% of the functional variance for all observations allocated to this state, based on investigation of the scree plot (S3 Fig). Eigenfunctions for State 1 are plotted in Fig 6.

The first functional principal component (FPC; red) explained 22.6% of the variance, and represents higher power concentrated in the delta (1.5 - 4 Hz) and beta-2 (22 – 30 Hz) ranges, contrasted with alpha (7 – 14 Hz) power and to a lesser extent theta (4-7 Hz) and beta-1 (14-22 Hz). The second FPC (orange) explained 17.1% of the variance and represents high relative beta power and moderate power for high-frequency alpha (10-14 Hz), contrasted with theta and low-frequency alpha (7-10 Hz), indicating that individuals with higher scores on this functional principal component tend to have a higher frequency alpha peak, low theta power, and a lower frequency beta peak. The third FPC (green) explained 13.6% of the variance and represents higher theta, alpha and beta-2 power contrasted with beta-1 power, with higher scores indicating greater alpha activity and higher frequency beta peaks relative to other individuals in this state. The fourth FPC (light blue) explained 11.5% of the variance and represents higher power in the theta, low-frequency alpha and beta-1 ranges contrasted with delta, high-frequency alpha and beta-2 power, indicating lower frequency alpha and beta

**Table 2. Demographics, cognitive function and psychopathology measures for individuals who spent 80% or more of their time in one state. Mean (SD).**

| Dominant State (> 80%) | State 1 (N = 167) | State 2 (N = 90) | State 3 (N = 52) | State 4 (N = 44) |
|---|---|---|---|---|
| **Demographics** | | | | |
| Female - N(%) | 59.0 (35.3%) | 30.0 (33.3%) | 16.0 (30.8%) | 14.0 (31.8%) |
| Age | 11.4 (1.6) | 11.0 (1.6) | 12.2 (1.6) | 12.1 (1.8) |
| EHQ | 62.5 (47.1) | 66.9 (39.7) | 64.5 (46.1) | 45.6 (60.8) |
| **Cognitive Function** | | | | |
| NIH Card | 93.0 (16.8) | 94.5 (16.5) | 95.8 (16.7) | 96.5 (17.7) |
| NIH Flanker | 85.9 (13.4) | 87.9 (14.8) | 89.1 (11.9) | 85.4 (10.8) |
| NIH List | 97.7 (15.4) | 97.8 (14.8) | 97.9 (13.2) | 99.5 (17.2) |
| NIH Pattern | 92.4 (21.7) | 94.3 (23.4) | 93.2 (22.1) | 93.7 (20.4) |
| **Psychopathology** | | | | |
| MFQ SR | 14.4 (10.3) | 14.5 (10.4) | 11.2 (10.5) | 15.3 (12.2) |
| SCARED SR | 23.3 (15.3) | 25.9 (17.0) | 20.2 (15.1) | 24.1 (17.5) |
| YSR Ext | 53.1 (11.3) | 52.4 (11.4) | 51.1 (10.5) | 51.0 (10.4) |
| YSR Int | 57.1 (11.9) | 57.8 (13.6) | 54.0 (11.6) | 58.5 (11.6) |

MFQ SR = Mood and Feelings Questionnaire, Self Report; SCARED SR = Screen for Child Anxiety Related Disorders, Self Report; YSR Ext = Youth Self Report, Externalising Scale; YSR Int = Youth Self Report, Internalising Scale; NIH = National Institutes of Health Toolbox Cognitive function tasks; NIH Card = Card Sorting task measuring executive function; NIH Flanker = Flanker task measuring executive function and attention; NIH List = List sorting task measuring working memory; NIH Pattern = Pattern comparison task measuring processing speed.

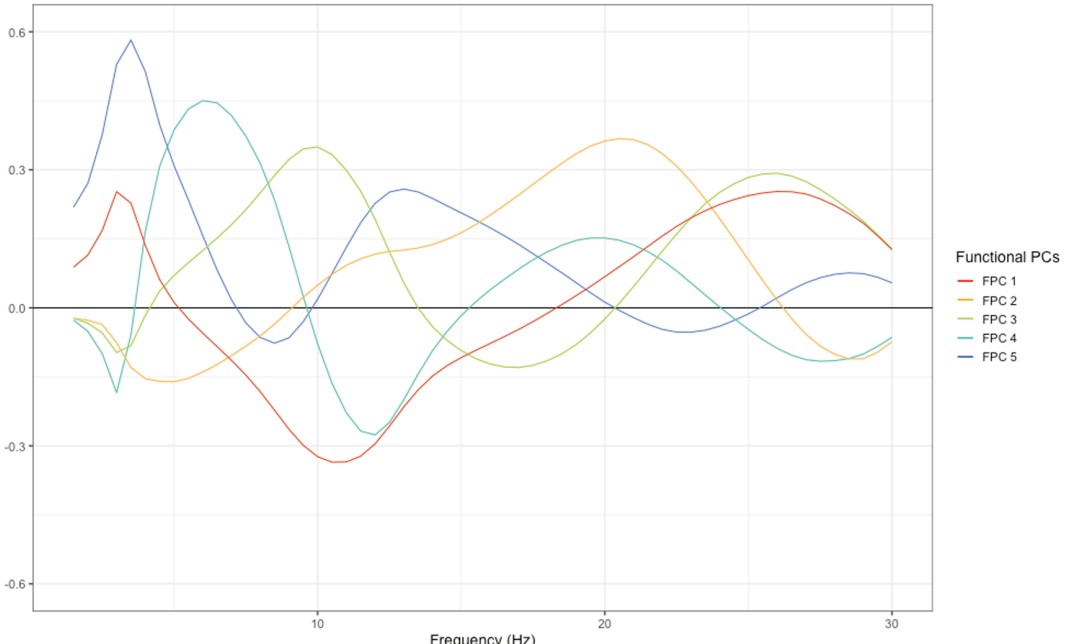

**Fig 6. Functional principal components (eigenfunctions) for observations allocated to State 1.**

peaks and low delta power. The fifth FPC (light blue) explained 7.8% of the variance and represents higher delta, theta and high-frequency alpha power. Similar to the second FPC, higher scores on the fifth FPC indicated higher frequency alpha peaks, but with a high amplitude delta peak and lower beta power.

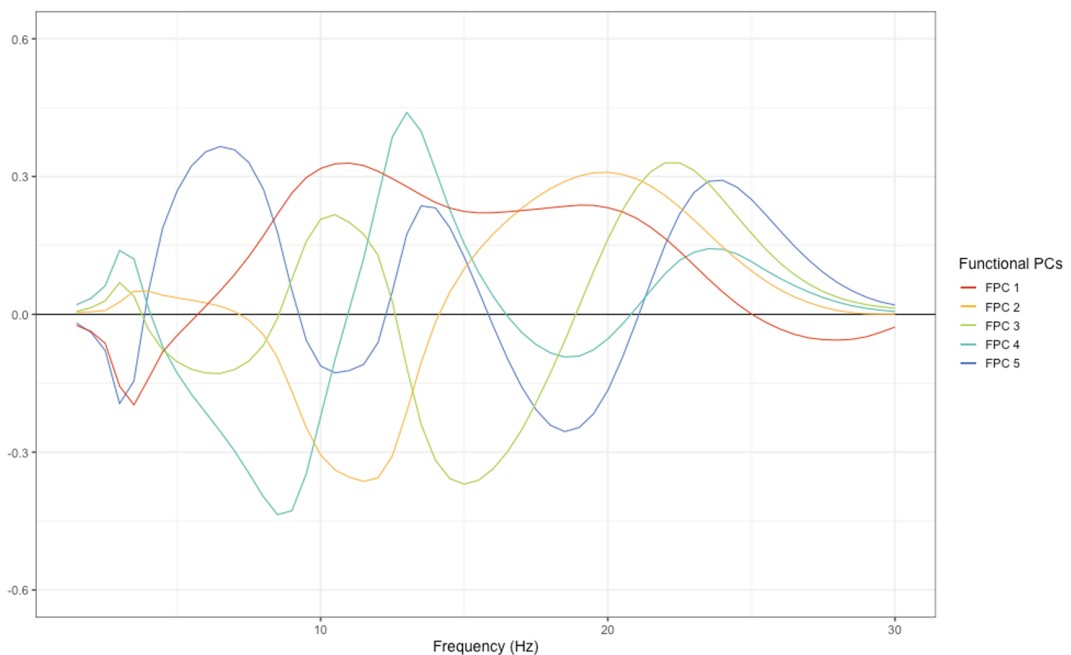

**Fig 7. Functional principal components (eigenfunctions) for observations allocated to State 3.**

**State 3 – FPCA.**  For State 3, we retained 5 functional principal components which cumulatively explained 77.4% of the functional variance for all observations allocated to this state, based on investigation of the scree plot (S5 Fig). Eigenfunctions for State 3 are plotted in Fig 7.

The first FPC explained 25.2% of the variance, and represents a high alpha and beta-1 power, contrasted with low delta and beta-2 power. The second FPC explained 24.3% of the variance and represents high frequency beta-1 power with a peak at 20 Hz, contrasted with low alpha power. The third FPC explained 13.9% of the variance and represents power at low-frequency alpha and beta-2, contrasted with theta, high-frequency alpha and beta-1 power. Among differences in other frequency ranges, scores on FPC3 indicate low frequency alpha peaks. The fourth FPC explained 10.9% of the variance and represents high power in the delta, very high frequency alpha and beta-2 ranges, contrasted with low power for theta, low frequency alpha and beta-1. Higher scores on FPC4 here indicate higher frequency alpha oscillations. The fifth FPC explained 6.7% of variance and represents high power for theta, upper alpha and beta-2, contrasted with power for delta, low-frequency alpha and beta-1.

## Bayesian regression models – Relating functional analysis characteristics and health measures

The Bayesian regression models revealed several substantial associations between functional analysis outputs and health measures. For State 1, FHMM dynamics showed the number of transitions was positively associated with NIH Flanker performance. FPCA weightings in State 1 showed negative associations between FPC3 and NIH Pattern performance, and between FPC4 and performance on NIH Card, Flanker and List tasks. For State 3, FHMM dynamics showed the number of states was positively associated with YSR Externalising, while number of transitions was negatively associated with YSR Internalising. Time spent in State 3 was negatively associated with SCARED SR and positively associated with NIH Pattern

performance. FPCA weightings in State 3 showed positive associations between FPC3 and both YSR scales (Externalising and Internalising), and between FPC4 and both SCARED SR and NIH Flanker performance. We encourage interested readers to compare these Bayesian regression results across Tables 3 and 4.

**State 1 – Bayesian regression model.** Table 3 presents regression coefficients and 95% credible intervals for the Bayesian regression model for State 1. This model revealed multiple substantial associations between functional analysis outputs from resting state brain activity and measures of psychopathology and cognitive function, while controlling for the effect of age, sex and handedness as covariates. Full regression results tables for all four states, including intercepts and regression coefficients for control covariates, are provided in the Supplementary Materials.

Among individuals for whom State 1 was dominant, the number of transitions between latent states was positively associated with performance on the NIH Flanker task measuring executive function and attention, with each additional transition associated with an average 6 points higher score on this task (95% CI [2.0, 9.9]). Scores on FPC3, indicating higher theta, alpha and beta-2 power contrasted with beta-1 power, were associated with poorer processing speed as measured by the NIH Pattern task ($\beta$ = -59 [-113, -4.6]). Weaker associations (with credible intervals spanning zero) were also found between FPC3 scores and the NIH Card ($\beta$ = -32 [-76, 11]) and List ($\beta$ = -33 [-71, 5.9]), measuring executive function and working memory. Scores on FPC4, indicating lower frequency alpha and beta peaks and low delta power, had substantial negative associations with all four cognitive function measures. Taken together these associations with FPC3 and FPC4 suggest that among individuals for whom State 1 is dominant, higher alpha and beta-2 power and lower delta power are broad features revealed by FPCA that are associated with poorer cognitive function.

Several weaker effects were also identified for higher anxiety levels, measured by the SCARED SR scale, being associated with number of states visited ($\beta$ = 8 [-3.3, 19]), percentage of time spent in dominant State 1 ($\beta$ = 35 [-5.1, 75]), and scores on FPC2 ($\beta$ = 16 [-1.7, 34]) indicating high power in the beta and high-frequency alpha ranges contrasted with theta power.

**State 3 – Bayesian regression model.** Table 4 presents regression coefficients and 95% credible intervals for the Bayesian regression model for State 3. In terms of temporal dynamics of latent state occupancy, the number of states visited was associated with higher externalising psychopathology on the YSR Externalising dimension ($\beta$ = 7.3 [0.11, 15]). Among individuals for whom State 3 was dominant, the number of transitions between states was associated with lower scores on YSR Internalising ($\beta$ = -4.5 [-7.7, -1.3]), YSR Externalising ($\beta$ = -2.5 [-5.5, 0.45]), anxiety symptoms on SCARED SR ($\beta$ = -3.9 [-8.1, 0.34]), and depressive symptoms on MFQ SR ($\beta$ = -2.8 [-5.8, 0.27]). The proportion of time spent in State 3 was associated with lower scores on SCARED SR ($\beta$ = -45 [-89, -1.1]), YSR Internalising ($\beta$ = -33 [-66, 0.03]), and higher scores on the NIH Pattern task ($\beta$ = 101 [15, 185]). Beyond the broad effect that individuals spending a majority of their time in State 3 had better cognitive function and lower psychopathology, an additional effect appears to be associated with moving frequently between State 3 and other states, as the number of transitions was associated with lower scores on all psychopathology measures.

Considering functional characteristics in the frequency domain, substantial associations with health measures were identified for scores on FPC3 and FPC4 in State 3. Scores on FPC3 indicate high power for low-frequency alpha and beta-2 contrasted with low theta and beta-1 power. Scores on FPC3 were associated with higher scores on YSR Internalising ($\beta$ = 24 [9.5, 38]), YSR Externalising ($\beta$ = 17 [3.9, 31]), and depressive symptoms on MFQ SR ($\beta$ = 13 [-0.76, 27]). Scores on FPC4 represent high power for high-frequency alpha and

**Table 3. Regression coefficients and 95% credible intervals from Bayesian regression model for State 1. Entries in bold indicate coefficients with 95% CI excluding zero.**

| Independent Variable | Beta | 95% Credible Interval | Dependent Variable |
|---|---|---|---|
| N. States | 1.7 | -6.2, 9.5 | MFQ SR |
| N. States | 8 | -3.3, 19 | SCARED SR |
| N. States | 3.7 | -4.3, 12 | YSR Ext |
| N. States | 2.4 | -6.1, 11 | YSR Int |
| N. States | -6.2 | -19, 6.5 | NIH Card |
| N. States | -9.2 | -20, 0.81 | NIH Flanker |
| N. States | -8.7 | -20, 3.0 | NIH List |
| N. States | 4.6 | -12, 21 | NIH Pattern |
| N. Transitions | 0.51 | -2.5, 3.6 | MFQ SR |
| N. Transitions | -1.1 | -5.4, 3.2 | SCARED SR |
| N. Transitions | -0.78 | -3.9, 2.4 | YSR Ext |
| N. Transitions | 0.33 | -3.0, 3.7 | YSR Int |
| N. Transitions | 3.2 | -1.8, 8.2 | NIH Card |
| **N. Transitions** | **6** | **2.0, 9.9** | **NIH Flanker** |
| N. Transitions | 2.3 | -2.2, 6.7 | NIH List |
| N. Transitions | -1.1 | -7.4, 5.2 | NIH Pattern |
| Dominant State % | 14 | -14, 43 | MFQ SR |
| Dominant State % | 35 | -5.1, 75 | SCARED SR |
| Dominant State % | 13 | -17, 42 | YSR Ext |
| Dominant State % | 21 | -11, 51 | YSR Int |
| Dominant State % | -19 | -65, 28 | NIH Card |
| Dominant State % | 14 | -23, 50 | NIH Flanker |
| Dominant State % | -29 | -70, 11 | NIH List |
| Dominant State % | -11 | -71, 48 | NIH Pattern |
| S1 FPC1 | -4.9 | -19, 9.6 | MFQ SR |
| S1 FPC1 | -4.2 | -25, 16 | SCARED SR |
| S1 FPC1 | -4.2 | -19, 11 | YSR Ext |
| S1 FPC1 | -9.7 | -26, 6.0 | YSR Int |
| S1 FPC1 | -13 | -36, 10 | NIH Card |
| S1 FPC1 | -12 | -30, 7.0 | NIH Flanker |
| S1 FPC1 | 3 | -18, 24 | NIH List |
| S1 FPC1 | -9.6 | -39, 20 | NIH Pattern |
| S1 FPC2 | 3.3 | -9.3, 16 | MFQ SR |
| S1 FPC2 | 16 | -1.7, 34 | SCARED SR |
| S1 FPC2 | -0.35 | -13, 12 | YSR Ext |
| S1 FPC2 | 1.7 | -12, 15 | YSR Int |
| S1 FPC2 | -13 | -34, 7.4 | NIH Card |
| S1 FPC2 | 7.4 | -8.6, 23 | NIH Flanker |
| S1 FPC2 | -10 | -28, 8.1 | NIH List |
| S1 FPC2 | 4.2 | -21, 30 | NIH Pattern |
| S1 FPC3 | -7.7 | -34, 19 | MFQ SR |
| S1 FPC3 | -3.8 | -42, 34 | SCARED SR |
| S1 FPC3 | 5.6 | -22, 33 | YSR Ext |
| S1 FPC3 | -6.2 | -35, 23 | YSR Int |
| S1 FPC3 | -32 | -76, 11 | NIH Card |
| S1 FPC3 | -16 | -49, 18 | NIH Flanker |
| S1 FPC3 | -33 | -71, 5.9 | NIH List |
| **S1 FPC3** | **-59** | **-113, -4.6** | **NIH Pattern** |
| S1 FPC4 | 1.7 | -14, 18 | MFQ SR |
| S1 FPC4 | -7.6 | -30, 14 | SCARED SR |
| S1 FPC4 | 4.2 | -12, 20 | YSR Ext |
| S1 FPC4 | 3.6 | -13, 21 | YSR Int |

(*Continued*)

**Table 3.** (Continued)

| S1 FPC4 | -31 | -57, -5.4 | NIH Card |
|---|---|---|---|
| **S1 FPC4** | **-25** | **-45, -4.7** | **NIH Flanker** |
| **S1 FPC4** | **-25** | **-48, -2.2** | **NIH List** |
| S1 FPC4 | -23 | -56, 9.4 | NIH Pattern |
| S1 FPC5 | 0.9 | -33, 35 | MFQ SR |
| S1 FPC5 | -30 | -78, 19 | SCARED SR |
| S1 FPC5 | 17 | -19, 53 | YSR Ext |
| S1 FPC5 | -0.73 | -38, 37 | YSR Int |
| S1 FPC5 | 8.3 | -47, 65 | NIH Card |
| S1 FPC5 | 5.3 | -38, 49 | NIH Flanker |
| S1 FPC5 | -21 | -70, 29 | NIH List |
| S1 FPC5 | -59 | -129, 11 | NIH Pattern |

S1 FPC1 = Functional principal component 1 for State 1; MFQ SR = Mood and Feelings Questionnaire, Self Report; SCARED SR = Screen for Child Anxiety Related Disorders, Self Report; YSR Ext = Youth Self Report, Externalising Scale; YSR Int = Youth Self Report, Internalising Scale; NIH = National Institutes of Health Toolbox Cognitive function tasks; NIH Card = Card Sorting task measuring executive function; NIH Flanker = Flanker task measuring executive function and attention; NIH List = List sorting task measuring working memory; NIH Pattern = Pattern comparison task measuring processing speed.

beta-2 contrasted with low delta power. Scores on FPC4 were associated with higher scores on SCARED SR ($\beta$ = 25 [0.22, 50]), YSR Externalising ($\beta$ = 17 [-1.1, 35]), and MFQ SR ($\beta$ = 16 [-1.3, 34]), as well as higher scores on the NIH Flanker task ($\beta$ = 24 [1.8, 48]). Taken together, these results indicate that among individuals for whom State 3 is dominant, higher alpha power, lower theta power, and higher frequency beta oscillations may be indicative of greater psychopathology.

## Discussion

In this paper we have introduced *flawless* analysis, a novel functional analysis framework with a nested model structure that incorporates functional latent states and temporal dynamics using a FHMM, and frequency characteristics stratified by those states using functional principal component analysis. Applying *flawless* analysis to time series of power spectral densities calculated from EEG data, we have made novel discoveries that build on previous work regarding data-driven phenotypes of resting state brain activity in young people.

In the case of resting state brain activity measured using EEG data, modelling each PSD as a curve or functional data observation offers deeper insights compared to extracting features of relative power in specific canonical frequency bands. It remains common practice in EEG research to characterise brain activity across individuals in terms of relative power contained within these legacy frequency bands [14,15]. However, the use of fixed EEG power bands to characterise brain activity across individuals has received substantial criticism [16,65]. Traditional frequency bands for EEG (e.g. delta, theta, alpha, beta) do not correspond to consistent functional groupings of brain activity across individuals, especially in the age range of early adolescence when the frequency content of brain activity is known to be rapidly evolving [66]. Making inferences about oscillatory activity based on relative power within fixed canonical bands is liable to conflate true differences in oscillatory power with a number of other physiological processes including shifts in oscillation centre frequency within or between individuals [64], or changes in the aperiodic exponent of the frequency distribution [16]. In contrast to the traditional approach, a functional analysis approach based on periodic spectral features extracted from EEG data allows us to cater for inter-individual differences in oscillatory peak frequencies, make fewer assumptions about frequency content falling within traditional

**Table 4. Regression coefficients and 95% credible intervals from Bayesian regression model for State 3. Entries in bold indicate coefficients with 95% CI excluding zero.**

| Independent Variable | Regression Coefficient | 95% Credible Interval | Dependent Variable |
|---|---|---|---|
| N. States | 4.1 | -3.1, 11 | MFQ SR |
| N. States | 3.5 | -6.5, 14 | SCARED SR |
| **N. States** | **7.3** | **0.11, 15** | **YSR Ext** |
| N. States | 2.4 | -5.1, 9.8 | YSR Int |
| N. States | -6.5 | -21, 7.6 | NIH Card |
| N. States | -3 | -12, 6.2 | NIH Flanker |
| N. States | 1.8 | -13, 16 | NIH List |
| N. States | 9.1 | -9.8, 28 | NIH Pattern |
| N. Transitions | -2.8 | -5.8, 0.27 | MFQ SR |
| N. Transitions | -3.9 | -8.1, 0.34 | SCARED SR |
| N. Transitions | -2.5 | -5.5, 0.45 | YSR Ext |
| **N. Transitions** | **-4.5** | **-7.7, -1.3** | **YSR Int** |
| N. Transitions | -2.9 | -8.9, 3.1 | NIH Card |
| N. Transitions | -1.9 | -5.8, 2.1 | NIH Flanker |
| N. Transitions | -2 | -8.2, 4.2 | NIH List |
| N. Transitions | 3.3 | -4.6, 11 | NIH Pattern |
| Dominant State % | -16 | -47, 16 | MFQ SR |
| **Dominant State %** | **-45** | **-89, -1.1** | **SCARED SR** |
| Dominant State % | -7.2 | -38, 24 | YSR Ext |
| Dominant State% | -33 | -66, 0.03 | YSR Int |
| Dominant State % | -42 | -105, 19 | NIH Card |
| Dominant State % | -30 | -70, 11 | NIH Flanker |
| Dominant State % | -17 | -81, 47 | NIH List |
| **Dominant State %** | **101** | **15, 185** | **NIH Pattern** |
| S3 FPC1 | 1.4 | -14, 17 | MFQ SR |
| S3 FPC1 | 8.4 | -13, 30 | SCARED SR |
| S3 FPC1 | 1 | -14, 16 | YSR Ext |
| S3 FPC1 | 0.23 | -16, 16 | YSR Int |
| S3 FPC1 | 11 | -18, 41 | NIH Card |
| S3 FPC1 | 0.94 | -19, 20 | NIH Flanker |
| S3 FPC1 | -27 | -58, 4.0 | NIH List |
| S3 FPC1 | -19 | -60, 21 | NIH Pattern |
| S3 FPC2 | -3 | -14, 8.2 | MFQ SR |
| S3 FPC2 | -5.3 | -21, 10 | SCARED SR |
| S3 FPC2 | -0.3 | -12, 11 | YSR Ext |
| S3 FPC2 | -0.6 | -12, 11 | YSR Int |
| S3 FPC2 | 3 | -19, 25 | NIH Card |
| S3 FPC2 | 3 | -11, 17 | NIH Flanker |
| S3 FPC2 | -4 | -26, 18 | NIH List |
| S3 FPC2 | -21 | -51, 8.2 | NIH Pattern |
| S3 FPC3 | 13 | -0.76, 27 | MFQ SR |
| S3 FPC3 | 5.9 | -13, 24 | SCARED SR |
| **S3 FPC3** | **17** | **3.9, 31** | **YSR Ext** |
| **S3 FPC3** | **24** | **9.5, 38** | **YSR Int** |
| S3 FPC3 | -19 | -46, 7.8 | NIH Card |
| S3 FPC3 | -12 | -30, 5.0 | NIH Flanker |
| S3 FPC3 | -0.71 | -28, 27 | NIH List |
| S3 FPC3 | -3 | -39, 34 | NIH Pattern |
| S3 FPC4 | 16 | -1.3, 34 | MFQ SR |

(*Continued*)

**Table 4**. (Continued)

| S3 FPC4 | 25 | 0.22, 50 | SCARED SR |
|---|---|---|---|
| S3 FPC4 | 17 | -1.1, 35 | YSR Ext |
| S3 FPC4 | 8.8 | -10, 27 | YSR Int |
| S3 FPC4 | 11 | -25, 46 | NIH Card |
| **S3 FPC4** | **24** | **1.8, 48** | **NIH Flanker** |
| S3 FPC4 | -2.1 | -38, 34 | NIH List |
| S3 FPC4 | -40 | -88, 8.1 | NIH Pattern |
| S3 FPC5 | -1.8 | -37, 33 | MFQ SR |
| S3 FPC5 | 12 | -36, 60 | SCARED SR |
| S3 FPC5 | 2 | -34, 37 | YSR Ext |
| S3 FPC5 | -0.29 | -37, 36 | YSR Int |
| S3 FPC5 | 2.1 | -68, 70 | NIH Card |
| S3 FPC5 | -11 | -57, 33 | NIH Flanker |
| S3 FPC5 | 28 | -42, 99 | NIH List |
| S3 FPC5 | 89 | -3.6, 182 | NIH Pattern |

S3 FPC1 = Functional principal component 1 for State 3; MFQ SR = Mood and Feelings Questionnaire, Self Report; SCARED SR = Screen for Child Anxiety Related Disorders, Self Report; YSR Ext = Youth Self Report, Externalising Scale; YSR Int = Youth Self Report, Internalising Scale; NIH = National Institutes of Health Toolbox Cognitive function tasks; NIH Card = Card Sorting task measuring executive function; NIH Flanker = Flanker task measuring executive function and attention; NIH List = List sorting task measuring working memory; NIH Pattern = Pattern comparison task measuring processing speed.

**Table 5. Symbols and definitions.**

| Symbol | Description |
|---|---|
| $X$ | Functional data array |
| $p$ | Individual |
| $k$ | Time |
| $t$ | Functional domain |
| $s$ | Latent state in FHMM |
| $N$ | Number of latent states in FHMM |
| $i = 1, ..., N$ | Index for latent states in FHMM |
| $j$ | Index for state $s_{i+1}$ in FHMM |
| $C$ | Number of components retained in FPCA |

power bands, and capture more nuance and novel characteristics of interest in the frequency domain.

Notable findings include identifying patterns of difference in psychopathology and cognitive function between groups of individuals who spent 80% or more of their time in each latent state. From the perspective of applied neuroscience and mental health research, we found similar patterns and extended on earlier findings regarding data-driven phenotypes of resting state brain activity in young people [67]. We have identified 4 functional latent states which young people in this sample occupied during eyes closed, resting state brain activity. At a high level, these states were associated with substantial variation in psychopathology and cognitive function between individuals for whom each latent state was dominant. Notably, individuals mainly allocated to State 1 had elevated risk for psychopathology and poorer cognitive function, and broadly this state was characterised by having high delta power, very low alpha power, and high relative beta-2 power. Individuals mainly allocated to State 3 exhibited a profile of lower psychopathology and better cognitive function, and this state broadly featured low theta power, a high frequency alpha peak, and moderate beta-1 power with a beta peak around 21 Hz.

Using FPCA stratified by latent state, and subsequent Bayesian regression models, we have also identified associations between temporal dynamics of latent state occupancy, frequency characteristics within states, and a variety of important health outcomes. Findings that stand out from these analyses include that scores on FPC3 and FPC4 in State 1, with prominent features including power in the theta, low-frequency alpha and beta-2 ranges, were associated with poorer cognitive function relative to other individuals for whom State 1 was dominant (Table 2). For State 3, scores on FPC3 and FPC4 were associated with higher scores on several psychopathology measures, indicating that lower theta power, higher alpha power, and higher frequency beta peaks may be indicative of greater psychopathology among individuals for whom State 3 was dominant.

Our regression models also revealed a number of contrasting effects of state occupancy dynamics between State 1 and State 3. In State 1, number of transitions between states was associated with improved executive function, and weakly associated with improved working memory. In State 3, number of transitions was associated with reduced psychopathology across all four measures. In State 1, proportion of time spent in dominant state was weakly associated with higher psychopathology and poorer cognitive function across several measures. In State 3, proportion of time spent in the dominant state was associated with lower anxiety symptoms and internalising psychopathology, and better processing speed. These results are preliminary, and while the mechanisms are currently unclear for the differences observed here in the associations of health measures with temporal dynamics of latent state occupancy, the contrasting effects between states displays the detailed characteristics from functional analyses that differentiate individuals within and between dominant latent states.

Our findings extend on previous research on EEG characteristics in neurodevelopmental conditions and psychopathology. For instance, the high delta and low alpha power observed in our State 1, associated with poorer cognitive function and high psychopathology, aligns with previous studies linking higher delta-alpha ratios to increased incidence of social anxiety in adolescents [75] and autism spectrum disorder in children [72], as well as poorer cognitive function following brain injury or stroke [73,74]. However, our functional approach reveals more detailed patterns of relationships between frequency characteristics and cognitive measures. For instance, the association we found between scores on FPC4 in State 1 and poorer cognitive function suggests that the interplay between theta, low-frequency alpha and beta power, indicating the presence of lower peak frequencies for alpha and beta activity, may be more informative than examining these bands in isolation, as is common in traditional EEG analyses. This highlights the potential of our functional approach to uncover more complex spectral signatures of cognitive function and psychopathology, going beyond simple ratios to consider the entire spectral profile and its dynamics over time.

Overall these results demonstrate unique insights that are available through the *flawless* analysis framework. They may be indicative of interaction effects of state occupancy dynamics which vary according to dominant state, and suggest the presence of further subgroups or phenotypes which could be identified on the basis of latent state dynamics and frequency characteristics within states. This demonstrates the unique insights available through our nested FDA framework relative to other analytical approaches. The centroids of functional latent states identified here bear a close resemblance to the average power spectral densities identified across 5 clusters in our previous work [67]. Extension of this work could pursue formal replication and comparison of *flawless* with the previous methodology used to identify resting state EEG phenotypes. These findings warrant further investigation, for which a possible approach could be clustering individuals based on *flawless* outputs to develop more detailed data-driven phenotypes of resting state brain activity.

Functional data analysis (FDA) methods can offer different analytical perspectives compared to multivariate parametric approaches with manually extracted features, particularly in their ability to consider the complete functional form of the data [3]. FDA is a recent area of growth in statistical research, and our work contributes a novel approach for nesting multiple FDA tools together to analyse temporal, latent state and frequency characteristics in time-varying functional data. By integrating these approaches in a nested structure, we are able to generate richer insights into functional and time-frequency characteristics of time-varying functional data than those available through using the individual methods in isolation. At the first stage, FHMM analysis provides insight into latent state characteristics and temporal dynamics of latent state occupancy. This stage also importantly enables the analysis to compare 'like with like' - by stratifying resting state brain activity data by allocation to latent states, we can apply FPCA to understanding the functional harmonics in the frequency domain that distinguish individuals while occupying matched latent states. Without this method, applying FPCA alone to time series of functional data over individuals would be conflating variation across multiple distinct latent states, and disregarding temporal dynamics.

Our nested approach differs from direct FTS dimension reduction methods like FSSA [18] in its analytical goals and structure. While FSSA provides an elegant mathematical framework for decomposing functional time series using trajectory operators and functional singular value decomposition, *flawless* prioritises interpretability through its nested structure of state-based decomposition followed by within-state functional analysis. This approach allows us to first identify distinct functional states and their temporal dynamics, then examine detailed functional characteristics within matched states. The separation of these analytical levels helps preserve interpretability for clinical applications, while still capturing complex patterns in both the temporal and frequency domains. While direct FTS methods may be more mathematically elegant or computationally efficient, the nested structure of flawless offers unique benefits for applied researchers seeking to understand both broad state-based patterns and nuanced functional variation within states.

Other functional analysis methods that have been developed to simultaneously model temporal and frequency domains together for neuroscientific data (e.g. [5,12,13]) tend to do so at the cost of substantially increased model complexity. These approaches simultaneously analyse neuroscientific data in multiple functional domains including spatial, temporal and frequency, and are able to capture complex inter-related patterns among these combined functional domains. However, the results of these models tend to be challenging to interpret in terms of influential characteristics in each of the domains of interest, which is an obstacle to utility for applied researchers or practitioners to integrate novel findings of functional analysis with their existing understanding of the applied subject matter. By combining functional methods that analyse these individual domains in a nested structure, our method enables analysis of inter-related patterns of features between the levels of latent state characteristics, temporal patterns of state occupancy, and functional frequency characteristics within states, while also generating specific insights at each of these levels of analysis. As we have shown, outputs at these levels be combined as inputs for multivariate statistical tools to understand their shared relationships to external outcomes – including, for our application, psychopathology and cognitive function in young people.

The present study had several limitations. It used a restricted frequency range of 1.5 - 30 Hz for the sake of interpretability, and for computational performance of the FOOOF algorithm. This approach may have excluded potentially relevant features of resting state EEG phenotypes that exist outside of this frequency range. While our focus on the Cz electrode provided valuable insights, we acknowledge that this single-electrode approach limits our ability to capture spatial patterns of brain activity. Related to these limitations, future

research could also expand the frequency range under investigation to incorporate low delta (0 - 1.5 Hz) and gamma (30+ Hz) ranges, which may reveal additional features and provide a more comprehensive understanding of resting state EEG phenotypes. Another avenue for future extension could include analysis of data from multiple EEG channels, capturing activity from different brain regions such as prefrontal and frontal activity, which may be more directly related to some of the cognitive and psychopathological measures we studied. This approach could enable the investigation of spatial patterns and functional connectivity in resting state EEG phenotypes, providing a more holistic understanding of brain function and the potential relationships with psychopathology and cognitive function.

Given the focus on FHMM and FPCA in this paper, extensions to the *flawless* framework could investigate sensitivity analysis and comparison with other functional analysis methods for studying latent state dynamics and functional data reduction. For this work we chose to use FPCA, as it is a widely used and well-established method for functional data reduction [2,3], making it an appropriate choice for this initial effort to combine functional latent state and data reduction methods in a nested framework. However, other approaches to functional dimensionality reduction are available, including functional factor analysis and other variations on the orthogonal rotation between components for FPCA [68,69]. For functional HMMs, to our knowledge there are no existing implementations of alternative latent state analyses such as Hidden Semi-Markov Models or Conditionally Autoregressive HMMs, which can account for more complex structures of temporal autocorrelation in time series data [70,71]. Future work could pursue implementation of alternative latent state analysis methods for use with functional data, and compare performance in the *flawless* framework with the FHMM approach we have used.

While our current approach for initialising the FHMM algorithm relies on visual identification and averaging of stable centroids, we acknowledge that an algorithmic method for distinguishing centroids could enhance the robustness of our approach. The implications of averaging centroids for initialisation include potential loss of some fine-grained distinctions between states, but in this case it helps to identify stable, consistent patterns across subsamples and improve the stability of algorithm performance. Future work could explore algorithmic methods for initial centroid identification, including a functional implementation of k-means++ [55], to further improve the reliability and reproducibility of our results.

The applied findings presented in this paper provide a foundation for developing sophisticated data-driven phenotypes based on resting state EEG data. Using the outputs of flawless *analysis*, future work could explore clustering analyses to develop more detailed phenotypes, capturing not only broad differences in the characteristics of latent states, but also the temporal dynamics of individual trajectories among states, and frequency characteristics within states. This approach, which embraces the complexity and richness of functional data from resting state EEG, has potential to contribute to our understanding of brain activity patterns and contribute to a more comprehensive characterisation of individual differences in EEG phenotypes and their associations with psychopathology and cognitive function.

The *flawless* analysis framework, initially developed in this context of modelling resting state EEG data, has the potential for application across a wide range of fields that involve functional data. In disciplines such as ecology and biomedical research, functional data often captures complex temporal and spatial patterns that are difficult to analyse with traditional statistical methods. By adapting *flawless* analysis to accommodate the specific requirements of different domains, researchers can leverage its capabilities to explore previously unidentified relationships, identify novel patterns, and ultimately contribute to the advancement of knowledge in their respective areas of study.

## Appendix – Notation reference table
## Supporting information

\*\***S1 File.** Supplementary Materials\*\* Supplementary materials for manuscript: "Functional analysis within latent states: A novel framework for analysing functional time series data" (PDF)

## Acknowledgements

We extend our heartfelt gratitude to the participants and their caregivers in the Healthy Brain Network study. We are grateful to the Child Mind Institute and the leaders of the Healthy Brain Network for their pioneering research and generous provision of such a valuable open-source dataset to the community.

## Author contributions

**Conceptualization:** Owen Forbes, Edgar Santos-Fernandez, Paul Pao-Yen Wu, Kerrie Mengersen.

**Data curation:** Owen Forbes.

**Formal analysis:** Owen Forbes.

**Investigation:** Owen Forbes, Edgar Santos-Fernandez.

**Methodology:** Owen Forbes, Edgar Santos-Fernandez, Paul Pao-Yen Wu, Kerrie Mengersen.

**Software:** Owen Forbes.

**Supervision:** Edgar Santos-Fernandez, Paul Pao-Yen Wu, Kerrie Mengersen.

**Visualization:** Owen Forbes.

**Writing – original draft:** Owen Forbes.

**Writing – review & editing:** Edgar Santos-Fernandez, Paul Pao-Yen Wu, Kerrie Mengersen.

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
