## [Decision Letter · Decision Letter 0]

22 Aug 2024

PONE-D-24-19155Functional analysis within latent states: A novel framework for analysing functional time series dataPLOS ONE

Dear Dr. Forbes,

Thank you for submitting your manuscript to PLOS ONE. After careful consideration, we feel that it has merit but does not fully meet PLOS ONE’s publication criteria as it currently stands. Therefore, we invite you to submit a revised version of the manuscript that addresses the points raised during the review process.

We look forward to receiving your revised manuscript.

Kind regards,

Yu Zhou

Academic Editor

PLOS ONE

Journal Requirements:

"Full list of funders: Australian Research Council Centre of Excellence for Mathematical and Statistical Frontiers, CE140100049, KM Statistical Society of Australia, PhD Scholarship, OF Queensland University of Technology, PhD Scholarship, OF International Biometrics Society, PhD Scholarship, OF"

4. Please ensure that you refer to Figure 6 in your text as, if accepted, production will need this reference to link the reader to the figure.

6. Please update your submission to use the PLOS LaTeX template. The template and more information on our requirements for LaTeX submissions can be found at http://journals.plos.org/plosone/s/latex.

Reviewers' comments:

Reviewer's Responses to Questions

**Comments to the Author**

1. Is the manuscript technically sound, and do the data support the conclusions?

Reviewer #1: Yes

Reviewer #2: Partly

2. Has the statistical analysis been performed appropriately and rigorously? 

Reviewer #1: Yes

Reviewer #2: N/A

3. Have the authors made all data underlying the findings in their manuscript fully available?

Reviewer #1: Yes

Reviewer #2: Yes

4. Is the manuscript presented in an intelligible fashion and written in standard English?

Reviewer #1: Yes

Reviewer #2: Yes

5. Review Comments to the Author

Reviewer #1: PONE-D-24-19155 Review

Functional analysis within latent states: A novel framework for analysing functional time series data

The authors propose a nested approach to analyse EEG signals from a single electrode, Cz, using functional data analysis versions of HMM and PCA, in a large dataset, N = 503, of early adolescents aged 9–15. The paper is well written, although sometimes repetitive and with occasional claims without substantiation. Although I question the wisdom of naming a method flawless, the approach is methodologically sound and should find its way eventually into mainstream neuroimaging. However, there are several points which need to be addressed before accepting the paper, and some recommendations that would make the paper and method more attractive to the neuroimaging community.

Comment 1 - major

The authors describe the results but provide no interpretation with respect to neuroscience. How, for example, do these findings align or not with EEG results from other studies with children (e.g. ADHD) or adults (HMM analysis)? How are the canonical frequency bands related to psychopathy and cognition, and do the findings in this paper agree or challenge previous findings? Without some interpretation of the results, the paper may end up being considered interesting but irrelevant for many neuroscience researchers who are faced with a multitude of different methods to apply to their data.

Comment 2 - major

The authors choose to analyse the EEG signals from Cz (supplementary motor area and sensorimotor cortex, motor coordination, planning and execution) but regress the results against psychopathology and cognitive measures, the latter which are associated with prefrontal and frontal cortex. Re-running the analysis with Fp1/Fp2 would provide a higher level of credibility of the results, or alternativey, providing an explanation why Cz would capture signals related to prefrontal/frontal activity.

Comment 3 - minor

Abstract - The flawless framework offers superior flexibility and nuance. This is unsubstantiated and unnecessary.

Comment 4 – minor

Page 3

Ref 11 – date is missing in the reference.

Ref 15 is a preprint and does not represent ‘common practice’ in EEG research.

Comment 5 – minor

Page 4

HMM … latent states – remove ‘dominant modes’ as these are relevant for FPCA.

Comment 6 – minor

Page 5 and others

The use of ‘longitudinal’ is incorrect. The method can be applied to any time-varying functional data.

Comment 7 – minor

Page 5

‘In this work… using regression of clustering approaches’. There is no methodology outlined in the paper for regression of clustering approaches.

Comment 8 – minor

Page 6 – repetition – remove ‘building a biobank’

Comment 9 – minor

Page 7 and page 32

‘predominantly allocated to each latent state’. ‘…mainly… This implies that individuals were allocated to multiple dominant states which I do not believe was the case.

Comment 10 – minor

Page 10 ‘clinical relevance’ – I believe this should be clinical utility i.e. it is easy to measure in the clinic.

‘choosing high quality measures’ – unsubstantiated.

Comment 11 – minor

Page 13 – Repetition – suggest removing ‘Differing from a …’

Comment 12 – minor

Page 14 – ‘Visual identification’ and ‘averaged’. The method would be more robust if you could find an algorithmic way to distinguish the centroids. What are the implications of averaging the centroids for initiation?

Comment 13 – minor

Page 15 – What criteria did you use to ‘assess’ the scree plots? Also ‘on inspection’ – for what?

Comment 14 – minor

Page 16 and elsewhere – Reference to section 3.1 etc. which do not exist.

Comment 15 – minor

Page 17 – Table 1 EHQ is not defined, and the order of the abbreviations does not follow the order in the table.

‘FOOOF performance….’ What is this and why is it here?

Comment 16 – minor

Page 18 and elsewhere – Figure 2 in the text should be Figure 3, etc for all subsequent figures.

Comment 18 – major

Page 19 – Section ‘Comparing Psychopathology….’ And Figure 5. I would like to see ANOVAs on the psycho and cognitive measures across the 4 dominant state groups. The statements on contrasts among the groups is not statistically substantiated.

Comment 19 – minor

Page 23 and elsewhere - ‘individuals with higher scores...’ What scores – do the authors mean the FPCA weightings?

Comment 20 – minor

Page 26-31 – I would suggest putting Table 3 and 4 together so that readers can compare the regression results across State1 and State 3 dominance groups.

Comment 21 – major

Page 26 – I would suggest putting a brief summary of the results at the start of the section ‘Bayesian Regression…’ highlighting that FHMM dynamics were statistically associated with NIH Flanker in the State 1 group, and YSR Ext, YSR Int, SCARED SR, and NIH Pattern in the State 3 group. With the additional FPCA analysis, individual weightings were statistically associated with NIH Pattern, NIH Card, NIH Flanker and NIH List in the State 1 group, and weightings in the State 3 group were statistically associated with YSR Ext, YSR Int, SCARED SR, and NIH Flanker.

These findings and the insights they provide should be discussed in the discussion section.

Comment 22 – minor

Page 32 – repetition – remove second paragraph of the discussion.

Comment 23 – major

Page 33 - ‘These regresions….’ This is a description of the results which should eb interpreted in the context of previous findings in EEG.

‘…FDA methods can offer…’ – the reference to support the claim comes from 2014 – and the claim is exaggerated.

Comment 24 – minor

Page 34 – ‘…decreased interpretability...’ – this is a critique of alternative methods, but the method in this paper lacks interpretation!

Comment 25 – minor

Page 35 – ‘… well validated’. Again, this statement is unsubstantiated. What type of validation are you referring to? Where are the citations for the validation?

Inconsistency – dimension reduction, dimensional reduction – dimensionality reduction is more commonly used in neuroscience.

‘… significantly enhance our understanding…’ The results are descriptive – maybe contribute to the understanding otherwise the statement is an exaggeration.

Reviewer #2: The manuscript titled “Functional analysis within latent states: A novel framework for analysing functional time series data” proposes a novel method “flawless” based on functional data analysis (FDA) of time series data. The proposed framework uses functional principal component analysis (FPCA) and functional hidden Markov model (fHMM) to identify latent states in EEG signals. The authors explore the relationship between the latent states and measures of psychopathology and cognitive function provided alongwith the EEG data. The authors approach to predict the latent states and their association with behavorial measures is interesting and of wide interest to the community. However I have some reservations about the manuscript mainly pertaining to the efficacy and presentation of the method.

In the introduction section of the manuscript, there are hardly any mention of fda based methods for exploring brain signals. Specifically, authors should include previous work in EEG signals based of FDA, FPCA and its variants. The authors should also include studies on EEG classification using HMM latent states. Finally, the literature review should also include latest methods and results using the EEG data from Healthy Brain Network the authors use in the study. The literature review section in the manuscript in general needs to be re-written.

The FDA portion of the method section is poorly written and requires more details. Given that this is primarily a method paper, the authors should elaborate on how a time series data is converted to a functional data and mention the assumptions and constraints underlying the functional data obtained from a time series. The authors should also specially mention that FDA works on only single channel and hence spatio-temporal latent states or signal analysis could not be performed with the proposed method. A pseudo code of the algorithm along with the framework as given in Figure 1 will also help the readers to better understand the steps involved in flawless and make it easier to implement.

I have the most reservation on the results sections. Although the authors mention in the abstract that “compared to traditional multivariate statistical methods, the flawless framework superior flexibility and nuance”, but I find no demonstration of the author’s claim. Although the proposed algorithm looks promising, the authors failed to show its superior performance as the flawless is not compared with any other method. The authors must demonstrate that indeed flawless performance is better than traditional methods by comparing it with traditional methods and demonstrating the results. The authors could demonstrate the method’s performance when using the same algorithm but using time series data instead of functional data. Since the authors are using EEG data from a public database, they have to additionally demonstrate the superior performance of the proposed method when compared with other recent methods used on the same data. Finally, along with comparisons with traditional multivariate methods, the authors should also compare with recent FDA based methods.

Overall the manuscript is interesting the manuscript is interesting and proposed framework looks efficient. However the manuscript needs improvement in the literature review section in the Introduction and the results using proposed methods needs to be compared with other state-of-the-art methods.

6. PLOS authors have the option to publish the peer review history of their article (what does this mean?). If published, this will include your full peer review and any attached files.

Reviewer #1: **Yes: **Fran Hancock

Reviewer #2: No

---

## [Author Response · Author response to Decision Letter 1]

23 Jan 2025

Dear Professor Yu Zhou,

Thank you for your positive decision on our manuscript. Please find our revised version attached. We believe that the manuscript is much improved and we are grateful to you and the reviewers for providing such useful feedback. We also thank both reviewers for carefully reading the manuscript and their constructive feedback.

Please find attached a detailed response to your comments. These reflect the amendments in the revised manuscript.

Kind regards,

Owen Forbes and colleagues

---

## [Decision Letter · Decision Letter 1]

31 Mar 2025

PONE-D-24-19155R1Functional analysis within latent states: A novel framework for analysing functional time series dataPLOS ONE

Dear Dr. Forbes,

Thank you for submitting your manuscript to PLOS ONE. After careful consideration, we feel that it has merit but does not fully meet PLOS ONE’s publication criteria as it currently stands. Therefore, we invite you to submit a revised version of the manuscript that addresses the points raised during the review process.

**ACADEMIC EDITOR:** Kindly address the pending comments. 

We look forward to receiving your revised manuscript.

Kind regards,

Academic Editor

PLOS ONE

Journal Requirements:

Reviewers' comments:

Reviewer's Responses to Questions

**Comments to the Author**

1. If the authors have adequately addressed your comments raised in a previous round of review and you feel that this manuscript is now acceptable for publication, you may indicate that here to bypass the “Comments to the Author” section, enter your conflict of interest statement in the “Confidential to Editor” section, and submit your "Accept" recommendation.

Reviewer #1: All comments have been addressed

Reviewer #3: (No Response)

2. Is the manuscript technically sound, and do the data support the conclusions?

Reviewer #1: (No Response)

Reviewer #3: Yes

3. Has the statistical analysis been performed appropriately and rigorously? 

Reviewer #1: (No Response)

Reviewer #3: Yes

4. Have the authors made all data underlying the findings in their manuscript fully available?

Reviewer #1: (No Response)

Reviewer #3: Yes

5. Is the manuscript presented in an intelligible fashion and written in standard English?

Reviewer #1: (No Response)

Reviewer #3: Yes

6. Review Comments to the Author

Reviewer #1: (No Response)

Reviewer #3: (No Response)

7. PLOS authors have the option to publish the peer review history of their article (what does this mean?). If published, this will include your full peer review and any attached files.

Reviewer #1: **Yes: **Fran Hancock

Reviewer #3: No

---

## [Author Response · Author response to Decision Letter 2]

5 May 2025

Dear Reviewer,

Thank you for your valuable feedback. We have responded to your suggestions in the attached rebuttal letter, and made changes accordingly in the attached manuscript files (clean and track changes versions).

Kind regards,

Owen Forbes

---

## [Decision Letter · Decision Letter 2]

3 Jun 2025

Functional analysis within latent states: A novel framework for analysing functional time series data

PONE-D-24-19155R2

Dear Dr. Forbes,

We’re pleased to inform you that your manuscript has been judged scientifically suitable for publication and will be formally accepted for publication once it meets all outstanding technical requirements.

Kind regards,

Hilary Izuchukwu Okagbue, Ph.D

Academic Editor

PLOS ONE

Additional Editor Comments (optional):

Reviewers' comments:

Reviewer's Responses to Questions

**Comments to the Author**

1. If the authors have adequately addressed your comments raised in a previous round of review and you feel that this manuscript is now acceptable for publication, you may indicate that here to bypass the “Comments to the Author” section, enter your conflict of interest statement in the “Confidential to Editor” section, and submit your "Accept" recommendation.

Reviewer #1: All comments have been addressed

Reviewer #3: All comments have been addressed

2. Is the manuscript technically sound, and do the data support the conclusions?

Reviewer #1: Yes

Reviewer #3: Yes

3. Has the statistical analysis been performed appropriately and rigorously? 

Reviewer #1: Yes

Reviewer #3: Yes

4. Have the authors made all data underlying the findings in their manuscript fully available?

Reviewer #1: Yes

Reviewer #3: Yes

5. Is the manuscript presented in an intelligible fashion and written in standard English?

Reviewer #1: Yes

Reviewer #3: Yes

6. Review Comments to the Author

Reviewer #1: (No Response)

Reviewer #3: (No Response)

7. PLOS authors have the option to publish the peer review history of their article (what does this mean?). If published, this will include your full peer review and any attached files.

Reviewer #1: **Yes: **Fran Hancock

Reviewer #3: No

---

## [Editor Report · Acceptance letter]

PONE-D-24-19155R2

PLOS ONE

Dear Dr. Forbes,

I'm pleased to inform you that your manuscript has been deemed suitable for publication in PLOS ONE. Congratulations! Your manuscript is now being handed over to our production team.

Kind regards,

on behalf of

Dr Hilary Izuchukwu Okagbue

Academic Editor

PLOS ONE